# Benchmarking of methods for DNA methylome deconvolution

Kobe De Ridder [1], Huiwen Che [1], Kaat Leroy [1] & Bernard Thienpont [1,2,3] ✉

Defining the number and abundance of different cell types in tissues is important for understanding disease mechanisms as well as for diagnostic and prognostic purposes. Typically, this is achieved by immunohistological analyses, cell sorting, or single-cell RNA-sequencing. Alternatively, cell-specific DNA methylome information can be leveraged to deconvolve cell fractions from a bulk DNA mixture. However, comprehensive benchmarking of deconvolution methods and modalities was not yet performed. Here we evaluate 16 deconvolution algorithms, developed either specifically for DNA methylome data or more generically. We assess the performance of these algorithms, and the effect of normalization methods, while modeling variables that impact deconvolution performance, including cell abundance, cell type similarity, reference panel size, method for methylome profiling (array or sequencing), and technical variation. We observe differences in algorithm performance depending on each these variables, emphasizing the need for tailoring deconvolution analyses. The complexity of the reference, marker selection method, number of marker loci and, for sequencing-based assays, sequencing depth have a marked influence on performance. By developing handles to select the optimal analysis configuration, we provide a valuable source of information for studies aiming to deconvolve array- or sequencing-based methylation data.

Profiling the number and abundance of different cell types in tissues is important for research into disease mechanisms as well as for diagnostic and prognostic purposes. For instance, the stromal cell composition of a tumor predicts response to therapy and patient survival, and provides insights into drug resistance mechanisms[1–3]. Typically, cell fractions in tissue are determined using immunohistological analyses, cell sorting, or single-cell RNA-sequencing. Though valuable in specific settings, these techniques are either expensive and time-consuming, or limited in sensitivity by a restrictive availability and multiplexing ability of highly specific antibodies[4–6]. Additionally, they are not applicable when querying the abundance of tissues- or cells-of-origin in a complex mixture of nucleic acids, as found for example in blood plasma under the form of cell-free DNA or RNA[7–12].

A valuable alternative is provided by techniques that determine the individual cell types contributing to a mixture, by decomposing the signal derived from this mixture into its constituent signals, a procedure called "deconvolution"[13–16]. Deconvolution is often applied to bulk transcriptomes, with expression of cell-type-specific marker genes reflecting the contribution of that cell type to the bulk profile. A plethora of deconvolution methods exist, and a recent benchmarking study on transcriptome deconvolution demonstrated that they display a variable performance[17]. Although transcriptomes can indeed reveal cell-type compositions, they are inherently variable between samples and individuals due to variations in RNA quality. Additionally, transcriptional activity is dependent on cell type. Different cell types hence contribute to a varying degree to the total RNA content of a bulk sample[18–20].

[1]Laboratory for Functional Epigenetics, Department of Human Genetics, KU Leuven, 3000 Leuven, Belgium. [2]KU Leuven Institute for Single Cell Omics (LISCO), KU Leuven, 3000 Leuven, Belgium. [3]KU Leuven Cancer Institute (LKI), KU Leuven, 3000 Leuven, Belgium. ✉e-mail: bernard.thienpont@kuleuven.be

**Table 1 | Overview of included deconvolution algorithms**

| Method | Statistical model | Software package |
|---|---|---|
| BVLS | Bounded-variable least squares | bvls R-package[28] |
| DCQ | Elastic net regularization | ADAPTS R-package[38] |
| Elastic net regression | Elastic net regularization | glmnet R-package[29] |
| EMeth-Binomial | Expectation maximization with binomial likelihood function | EMeth R-package[30] |
| EMeth-Laplace | Expectation maximization with Laplace likelihood function | EMeth R-package[30] |
| EMeth-Normal | Expectation maximization with normal likelihood function | EMeth R-package[30] |
| EpiDISH | Robust partial correlation | EpiDISH R-package[31] |
| FARDEEP | Least trimmed squares | FARDEEP R-package[36] |
| ICeDT | Expectation maximization | ICeDT R-package[37] |
| Lasso | Lasso regularization | glmnet R-package[29] |
| Meth atlas | Non-negative least squares | Meth atlas Python-package[32] |
| MethylResolver | Least trimmed squares | MethylResolver R-package[33] |
| Minfi | Linear constrained projection | Minfi R-package[34] |
| NNLS | Non-negative least squares | nnls R-package[35] |
| OLS | Ordinary least squares | lm base R function |
| Ridge | Ridge regularization | glmnet R-package[29] |

DNA methylomes are a frequently used alternative data source. Like gene expression, DNA methylation patterns are cell-type specific and amenable to high-throughput profiling[21–25]. They however possess a few advantages for deconvolution. Firstly, similar to marker genes, differential methylation at selected CpGs can serve as a cell-type-specific marker, but there are vastly more CpGs than expressed genes that can be used for deconvolution. Secondly, DNA methylation is binary, being either present or absent at a given locus in contrast to the continuous distribution of RNA transcription, rendering deconvolution more straightforward. Thirdly, assuming ploidy is comparable between the cell types, each cell will provide an equal contribution to the mixture, in contrast to the transcriptional magnitude-dependent contribution of cells in bulk transcriptomes, with some cell types containing orders of magnitude more RNA than others. A few studies on benchmarking of reference-based DNA methylation deconvolution have been described. These compare a limited number (4–6) of reference-based deconvolution methods and assess a limited set of variables that may impact deconvolution performance[26,27].

We here comprehensively evaluate 16 deconvolution algorithms, developed either specifically for DNA methylome data or for transcriptome data but having a generic basis. Different normalization methods are applied and tested on array- and sequencing-based DNA methylation profiles. We also assess the impact of other variables that may influence deconvolution performance, including the cellular fraction, the method of marker selection, the number of markers used to build the reference, the impact of technical variability, and the depth and evenness of sequencing. Together, these analyses allow tailored selection of methods for accurate DNA methylome deconvolution.

## Results
### Setting up the benchmarking
Methods to deconvolve DNA methylome profiles of heterogeneous samples into their constituent cell or tissue types can be broadly categorized into linear methods and more complex machine learning models. For our benchmarking, we selected 16 commonly used or recently developed methods that leverage a range of statistical algorithms, including expectation maximization, (regularized) least squares regression, robust partial correlation, and linear constrained projection. These methods are *BVLS, Elastic net regression, EMeth-Binomial, EMeth-Laplace, EMeth-Normal, EpiDISH, Lasso, Meth atlas, MethylResolver, Minfi, NNLS, OLS, Ridge regression, FARDEEP, ICeDT* and

*DCQ* (Table 1)[28–38]. We opted not to test reference-free methods such as *PRISM, MethylPurify, DXM,* and *csmFinder + coMethy*, which were benchmarked recently elsewhere and which have a performance inferior to reference-based methods[26]. Since many of these methods have different underlying assumptions of data structure and distribution, we also tested seven data normalization approaches (Supplementary Data 1), summing up to 112. Since *ICeDT* cannot deal with negative values, it is incompatible with (column) Z-score and log normalization, resulting in a total of 109 combinations.

Deconvolution is typically performed on a limited subset of marker loci. For most analyses, we used a fixed number of markers per cell type ($n = 100$ for each source) such that each cell or tissue type had an equal representation in the reference. Markers were selected essentially as described by Luo et al.[39] with a minor modification (see methods)[39]. To deconvolve mixtures of four tissue types, we thus identified 400 marker loci, while we identified 600 marker loci for mixtures of six leukocyte types (Supplementary Data 2). The same set of loci was used for every comparison. As a ground truth, 200 in silico mixtures were generated by combining single DNA methylation profiles of defined tissues or cell types (Supplementary Data 3) in specified proportions. Individual fractions were sampled from a uniform univariate distribution ranging between 0 and 1, after which values were rescaled to add up to 1. This resulted in homogeneous proportion distributions allowing us to compare performance between cell and tissue types in an unbiased manner. For each in silico mixture, methylation signals for each cell type were sampled from one randomly selected sample in the set of DNA methylome profiles available for that cell type, to reflect technical variation in samples between mixtures. We provide a detailed description of all in silico mixtures in Supplementary Data 3. Furthermore, to evaluate performance on real-life datasets, we included both in vitro mixed and cytometry-quantified whole-blood datasets that were previously described (see methods).

We tested the performance of each algorithm-normalization combination by computing measures of accuracy between deconvolved and actual proportions. We assessed deconvolution performance by quantifying the root mean square error (RMSE), reflecting the absolute error between true and predicted values, the Spearman's $R^2$, which quantifies the correlation between true and predicted values but is less perceptive of systematic biases, and Jensen–Shannon divergence (JSD), a performance metric assessing homogeneity between predicted and actual fraction distributions. These metrics were compiled into a summary metric that we call the

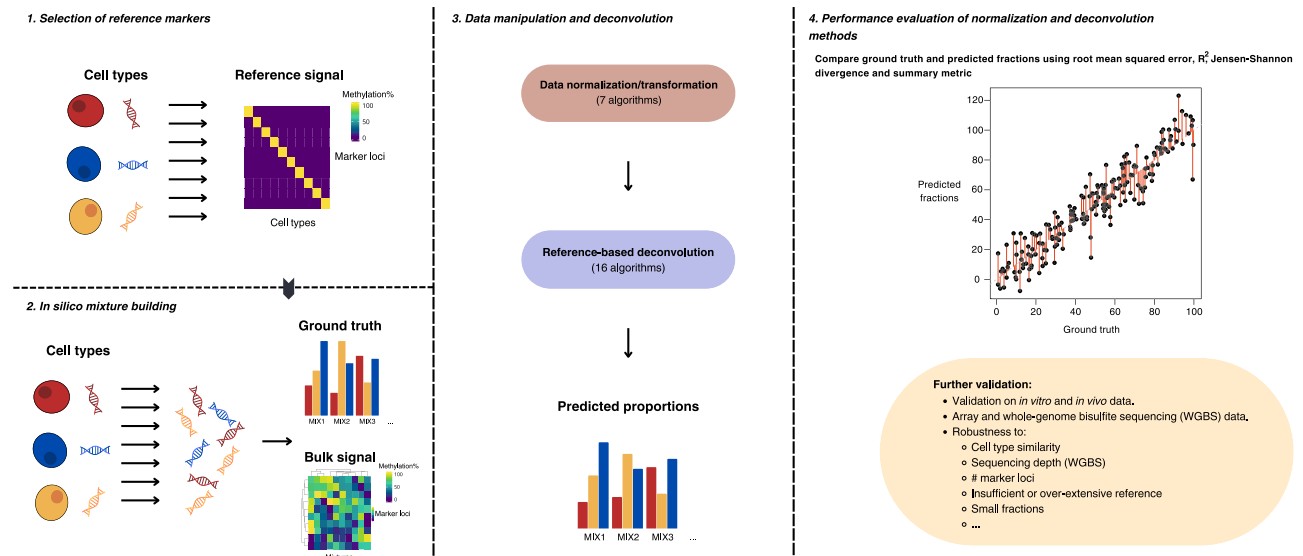

**Fig. 1 | Schematic representation of benchmarking workflow.** Panel **1**: selection of cell-specific markers using a reference dataset. Panel **2**: construction of in silico mixtures from an independent validation dataset. Panel **3**: normalization and deconvolution of bulk DNA methylation signals derived from in silico mixtures. Panel **4**: deconvolution performance assessment based on root mean squared error, Spearman's $R^2$, Jensen–Shannon divergence, and combined summary metric.

accuracy score (AS), and that combines the ranks of $R^2$, RMSE, and JSD (see methods). Practically, an RMSE difference of 0.01 represents an average absolute difference of 1% between predicted fractions and actual fractions. Some researchers may be more interested in relative differences of cell proportions between conditions and may thus consider $R^2$ to be more informative. We refer to Supplementary Data 4, 5, 6, and 7 for respectively the $R^2$, RMSE, JSD, and AS values of the various analyses.

When we applied deconvolution on in silico mixtures that were generated from the same dataset that serves as deconvolution reference, most methods produced near-perfect results (Supplementary Fig. 1) as expected. This differs from a real-world scenario, where reference profiles are generated from different samples, and often also profiled in different laboratories. To evaluate the real-world usage, we therefore selected separate, independently generated, reference datasets from the same cell or tissue sources to be used for marker selection and reference building (Fig. 1). These datasets differ markedly from those we use to generate our in silico mixtures, as was evident in a head-to-head comparison of DNA methylation levels at 100 marker loci (Fig. 2a, Supplementary Fig. 2a).

**Tissue fraction deconvolution**

As a first means of benchmarking all algorithm-normalization combinations, we focused on a relatively straightforward deconvolution problem, by assessing the deconvolution performance for mixtures of four tissues: small intestine (fraction range: 0.04%–70.76%), blood (fraction range: 0.54%–82.42%), kidney (fraction range: 0.07%–67.63%) and liver (fraction range: 0.05%–65.7%) (Supplementary Fig. 2), profiled using 450 K microarrays (HumanMethylation450K BeadChips; Illumina). The tissue types profiled vary in the specificity of the marker CpGs identified from the reference samples. This can be quantified by computing F-statistics for all cell types at their respective marker loci. Median values per cell type ranged from 125.5 for small intestine to 2045.3 for liver (Fig. 2b). Specificity of marker CpGs was also evident in the dataset used for in silico mixture generation, ranging from 173.3 for blood to 939.8 for liver (Fig. 2a, b). Indeed, although tissue type fractions were in general accurately predicted (median RMSE = 0.07, median Spearman's $R^2$ = 0.90), we observed significantly higher absolute error in small intestine (difference in RMSE = 0.10, $P < 10^{-16}$, CI 95%: 0.09, 0.10). Surprisingly however, significantly higher correlation was

observed between predicted and actual proportions for small intestine (difference in $R^2$ = 0.06, $P < 10^{-16}$, CI 95%: 0.06, 0.07). Though markers are highly specific in the dataset used to generate in silico mixtures, the dataset used for marker selection is noisier, resulting in a discordance between both (Supplementary Fig. 2a). This resulted in a consistent overestimation of small intestine proportions (Fig. 2c and Supplementary Fig. 2b), emphasizing the need to use high-quality datasets for achieving optimal deconvolution performance.

Disregarding the effect of normalization, *DCQ* (difference in AS to all other algorithms = −57.2; $P = 3.17 × 10^{-6}$, CI 95%: −90.0, −40.0) and *EMeth-Laplace* (difference in AS to all other algorithms = 54.5; $P = 0.004$, CI 95%: 14.0, 77.0) were the poorest and best-performing deconvolution methods respectively. Most normalization methods performed comparably, except for log transformation performing significantly worse than all others (difference in AS to all other normalizations = −115; $P < 10^{-16}$, CI 95%: −131.7, −90.0; Fig. 2d, e). The best prediction was achieved when combining *EMeth-Normal* deconvolution on column Z-score normalized data (median RMSE = 0.06, median Spearman's $R^2$ = 0.93, median AS = 294), while the worst prediction was produced by applying *EMeth-Normal* deconvolution on log normalized data (median RMSE = 0.19, median Spearman's $R^2$ = 0.84, AS = 54.8) (Fig. 2e).

**Impact of marker selection**

Effective marker selection is a major contributor to deconvolution performance. As described in the previous chapter, we selected the top 100 markers per cell type. Marker selection was based on both significance of FDR-adjusted $P$-values as well as on mean methylation differences between cell types, in line with a method proposed earlier (see methods)[39]. This allowed us to both select highly specific loci while simultaneously ensuring equal contribution of all cell types to the reference. Though beneficial for benchmarking purposes, this might compromise deconvolution performance. Therefore, we quantified how marker selection affects deconvolution performance by comparing our custom algorithm to the commonly used IDOL algorithm (43/400 marker loci overlapped between both sets)[40]. Generally, IDOL selection performed significantly worse, both in terms of $R^2$ (difference in $R^2$ = 0.02, $P < 10^{-10}$, CI 95%: 0.01, 0.03) and RMSE (difference in RMSE = −0.02, $P < 10^{-16}$, CI 95%: −0.03, −0.02) (Fig. 2f). This observation is in line with a previous benchmarking, which highlighted

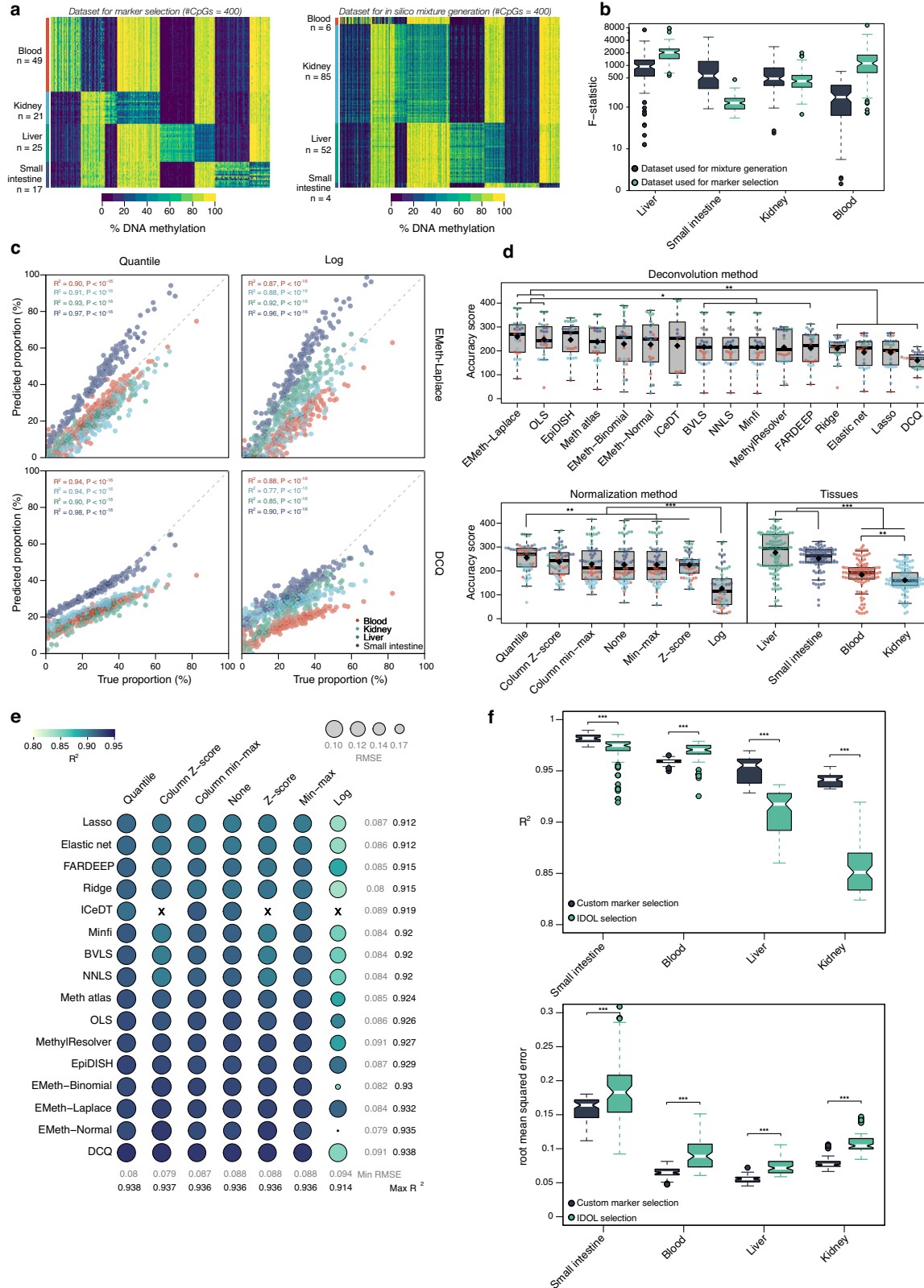

the same marker selection method as optimal when compared to five other methods.

## Impact of cell-type similarity

Next, we investigated how these algorithms perform a more difficult and relevant deconvolution task, namely the deconvolution of relatively homogeneous cell-type fractions that have a common

developmental origin. Specifically, we investigated how blood cell types can be reliably deconvolved, by constructing 200 pseudo-bulks generated from 450 K microarray profiles of fluorescence-activated cell sorting (*FACS*)-purified neutrophils (fraction range: 0.07%–51.74%), monocytes (fraction range: 0.07%–44.25%), CD4+ (fraction range: 0.07%–47.89%) and CD8+ T-cells (fraction range: 0.18%–45.88%), natural killer cells (fraction range: 0.28%–51.59%) and B-cells (fraction

**Fig. 2 | Deconvolution of tissues on Illumina 450 K array. a** Matrices of marker CpGs ($n = 400$) used for building the tissue methylation reference (left panel) and in silico mixtures (right panel) of 450 K data. Samples for four tissues were included: blood (reference: $n = 49$, validation: $n = 6$), kidney (reference: $n = 21$, validation: $n = 85$), liver (reference: $n = 25$, validation: $n = 52$), and small intestine (reference: $n = 17$, validation: $n = 4$). **b** Boxplots showing F-statistics for tissues at their respective marker CpGs, between reference ($n = 112$ biologically independent samples) and validation datasets ($n = 147$ biologically independent samples). The boxplots present median values and quartiles, whiskers the minimum and maximum values, and dots the individual data points. **c** Scatter plots showing true proportions (x-axis) and predicted proportions (y-axis) in percentages for all tissues for the best-performing (left-upper) and worst-performing (right-lower) algorithm-normalization combinations on 200 in silico mixtures. $R^2$ and p-values were calculated using Spearman's rank correlation test. **d** Deconvolution accuracy represented as boxplots showing accuracy scores for the different deconvolution methods, normalization methods, and cell types on 200 in silico mixtures. Black diamond shapes represent median

values, colors represent tissues. The boxplots present median values and quartiles, whiskers the minimum and maximum values, and dots the individual data points. P-values were determined using two-tailed FDR-adjusted Dunn's tests. **e** Performance of deconvolution on 200 in silico mixtures. Algorithm-normalization combinations are visualized as circles. Spearman's $R^2$ is represented by color, and root mean squared error is represented by size. Rows show deconvolution algorithms, columns show normalization methods. **f** Boxplots showing Spearman's $R^2$ and RMSE values for deconvolution on 200 in silico mixtures using markers selected by either custom or IDOL algorithm. The boxplots present median values and quartiles, whiskers the minimum and maximum values, and dots the individual data points. P-values were calculated using a two-tailed Wilcoxon rank-sum test. P-values for R²: small intestine = $5.2 \times 10^{-14}$; blood = $2.2 \times 10^{-16}$; liver = $2.2 \times 10^{-16}$; kidney = $2.2 \times 10^{-16}$. P-values for RMSE: small intestine = $4.2 \times 10^{-6}$; blood = $1.9 \times 10^{-15}$; liver = $2.2 \times 10^{-16}$; kidney = $2.2 \times 10^{-16}$. 'X' symbol represents missing values. Source data are provided as a Source Data file. Exact p-values are added in the Source Data file.

range: 0.3%–41.60%), and applying the same strategy we described earlier. 100 marker CpGs were identified for each cell type, resulting in a total of 600 loci (Fig. 3a and Supplementary Fig. 3a). Overall, algorithms showed similar performance (Fig. 3b). *EpiDISH* was overall the best-performing algorithm, significantly outperforming DCQ (difference in AS = 155.1, $P = 6.941 \times 10^{-5}$, CI 95%: 74.7, 211.7).

In terms of normalization, no methods significantly improved deconvolution over non-normalized data. Furthermore, log normalization resulted in significantly worse predictions (difference in AS to other normalizations = $-133$, $P = 1.422 \times 10^{-11}$, CI 95%: $-163$, $-100$; Fig. 3b–d). Notably, deconvolution was more accurate for some cell types than for others: indeed, quantification of natural killer and CD8+ T-cell abundance was poorer (difference in AS to other cell types = $-272.333$, $P < 10^{-16}$, CI 95%: $-288.3$, $-257.3$), perhaps because they are both cytotoxic effector cells thus sharing functional activities, despite originating from divergent lineages. Furthermore, DNA methylation for natural killer cell loci differed significantly between datasets used for marker selection and those used for mixture generation (difference in F-statistic = 91.18, $P < 10^{-16}$, CI 95%: 73, 110) suggesting technical variability between datasets (Supplementary Fig. 3b–d).

To further validate these rankings, we evaluated five DNA methylome profiles from whole-blood samples[41], comparing cell-type fractions predicted by deconvolution to cell-type fractions measured by flow cytometry and in vitro mixed blood samples. Overall deconvolution performance was lower, perhaps reflecting the additional measurement uncertainty introduced by applying flow cytometry (Fig. 3e, f). Though sample sizes were too small for statistical interpretation, *EpiDISH* was able to identify relative proportion differences well for most cell types (median $R^2 = 0.93$).

## Larger array size improves deconvolution

We next assessed the impact of array size on deconvolution efficiency, by analyzing DNA methylome profiles generated for the same cell types using EPIC microarrays (Infinium MethylationEPIC v1.0 BeadChip; Illumina), which encompass over 850,000 probes. These include most probes represented on 450 K microarrays, as well as about 350,000 additional probes targeting more enhancer CpGs and fewer CpG island CpGs than on the 450 K microarray[42]. Of note, all 600 marker CpGs we identified from 450 K arrays were also present on the EPIC arrays. 200 pseudo-bulks were again produced for neutrophils (fraction range: 0.5%–37.11%), monocytes (fraction range: 0.5%–54.70%), CD4+ (fraction range: 0.07%–42.37%) and CD8+ T-cells (fraction range: 0.04%–49.18%), natural killer cells (fraction range: 0.09%–43.98%) and B-cells (fraction range: 0.24%–48.73%), When using the same marker CpGs identified from 450 K data on EPIC array data, the performance for all 109 algorithm-normalization combinations improved significantly (difference in $R^2 = 0.04$; $P < 10^{-16}$, CI 95%: 0.040, 0.045), suggesting a higher concordance between the EPIC

array data used for generating in silico mixtures and for deconvolution (Supplementary Fig. 4).

We next identified 600 new marker CpGs from the EPIC array data. 484 CpGs were selected that do not overlap with those represented on the 450 K array (Fig. 4a and Supplementary Fig. 5a). Of these 484 additional marker CpGs, 246 overlap with known enhancer regions. As before, natural killer cells and CD8+ T-cells were not separated as accurately as other cell types (Fig. 4b). Lower cell specificity between these cell types was confirmed by a higher inter-sample correlation of CD4+ and CD8+ T-cells ($R^2 = 0.1$–$0.2$) as well as CD8+ and natural killer cells ($R^2 = 0.0$–$0.1$) compared to other cell-type pairs ($R^2 = 0.0$) (Supplementary Fig. 5b). Concordance in methylation ratios between datasets used for markers selection and generation of in silico mixtures was relatively high (Supplementary Fig. 5c, d). The relative performance rankings of algorithms and normalization methods were nevertheless comparable to those in the analyses using 450 K array probes described earlier (Fig. 4c). However, when comparing deconvolution using marker loci selected from the EPIC probes to those selected from 450 K array data, fraction estimates improved slightly for all cell types, except for CD8+ T-cells (Fig. 4d). To further validate these in silico analyses, we next assessed performances on DNA extracted from different cell types, mixed in vitro at prespecified ratios ($n = 12$)[43]. Here, deconvolution performance was comparable to the in silico generated mixtures, thus validating the relative rankings of deconvolution and normalization methods, as well as our strategy for generating pseudo-bulks (Fig. 4d–f).

## Impact of technical and biological variation on deconvolution

The above analyses represent a realistic scenario wherein an independently generated dataset is used as reference for bulk deconvolution. This approach however likely underestimates deconvolution performances, given the confounding technical and biological differences between reference and test, with variations in age, smoking, BMI or sex having a known impact on DNA methylomes[43]. To gauge their impact, we next removed this inter-dataset variability, by splitting a single dataset into a subset for reference generation and marker selection, while the remainder was used for in silico mixture generation. As expected, this improved the overall deconvolution performance (Supplementary Fig. 6a, b). Nevertheless, relative performances of deconvolution methods were comparable (Supplementary Fig. 6c). Furthermore, we investigated the effect of age differences between the dataset used for both reference generation and marker selection and the dataset used for mixture generation. We either built the reference using data from young individuals (<30 years old) and 200 in silico mixtures using data from older individuals (> = 30 years old), or the other way around. Though in general deconvolution performance was worse compared to deconvolving without age differences between reference and mixtures, the relative ranking of methods was highly similar (Supplementary Fig. 6d, e).

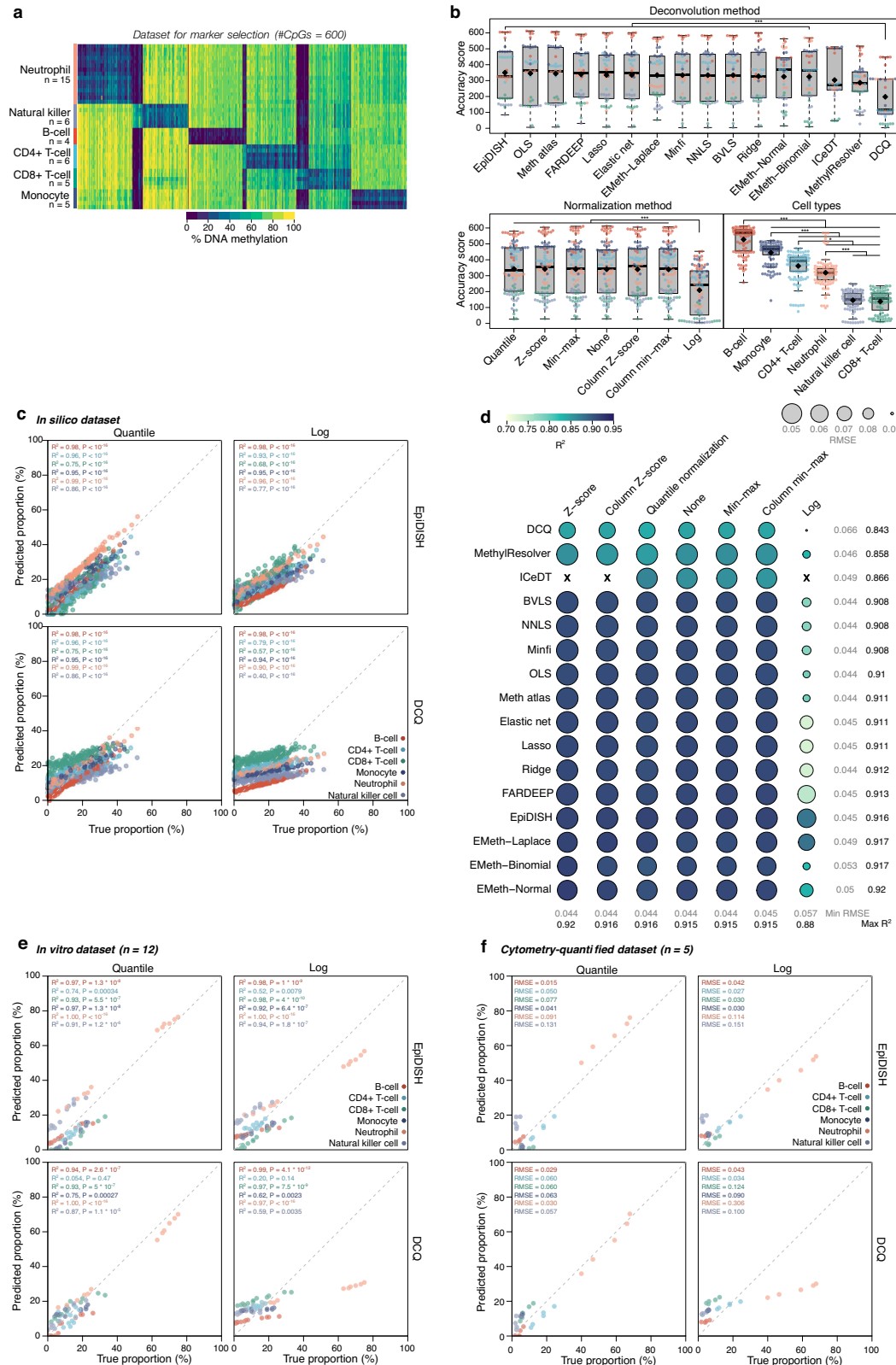

These results emphasize the need to select reference data that match the test cohort as much as possible but also validate our method rankings across technical and biological variation.

## Impact of the number of marker CpGs on deconvolution

All analyses described earlier rely on 100 marker CpGs per cell or tissue type. Depending on the method used to analyze DNA methylation, a lower number of marker CpGs may be preferable (e.g., when the cost is to be minimized). To assess the impact of the number of marker loci on deconvolution, we next repeated our performance assessment, while varying the number of marker CpGs included. Specifically, we selected the 2–500 top-ranked marker CpGs per cell type and assessed performance for each algorithm on unnormalized data (normalization did not improve performance; Fig. 5a, b). For most algorithms, the

**Fig. 3 | Deconvolution of immune cell types on Illumina 450 K array. a** Matrix of marker CpGs ($n = 600$) used for building immune cell methylation reference of 450 K data. Samples for six cell types were included: neutrophil ($n = 15$), natural killer cell ($n = 6$), B-cell ($n = 4$), CD4+ T-cell ($n = 6$), CD8+ T-cell ($n = 5$) and monocyte ($n = 5$). **b** Deconvolution accuracy represented as boxplots showing accuracy scores for the different deconvolution methods, normalization methods, and cell types on 200 in silico mixtures. Black diamond shapes represent median values, colors represent cell types. The boxplots present median values and quartiles, whiskers the minimum and maximum values, and dots the individual data points. *P*-values were determined using two-tailed FDR-adjusted Dunn's tests. *$P < 0.05$, **$P < 0.01$, ***$P < 0.001$. **c** Scatter plots showing true proportions (x-axis) and predicted proportions (y-axis) in percentages for the best-performing (left-upper) and worst-performing (right-lower) algorithm-normalization combinations on 200 in silico mixtures. $R^2$ and *p*-values were calculated using Spearman's rank correlation test.

**d** Performance of deconvolution on 200 in silico mixtures for all algorithm-normalization combinations represented as circles. Spearman's $R^2$ is represented by color, root mean squared error is represented by size. Rows show deconvolution algorithms, columns show normalization methods. **e** Scatter plots showing true proportions (x-axis) and predicted proportions (y-axis) in percentages for the best-performing (left-upper) and worst-performing (right-lower) algorithm-normalization combinations on 12 in vitro mixtures. $R^2$ and *p*-values were calculated using Spearman's rank correlation test. **f** Scatter plots showing true proportions (x-axis) and predicted proportions (y-axis) in percentages for the best-performing (left-upper) and worst-performing (right-lower) algorithm-normalization combinations on five FACS-sorted whole-blood samples. $R^2$ and *p*-values were calculated using Spearman's rank correlation test. 'X' symbol represents missing values. Source data are provided as a Source Data file. Exact *p*-values are added in the Source Data file.

performance increased starting from two CpGs per cell type (median $R^2 = 0.97$) consistently when more marker CpGs were included, reaching an optimum at 75–100 CpGs (median $R^2 = 0.99$). This explains our choice of the number of markers for benchmarking. Both *EMeth-Binomial* and *EMeth-Normal*, reached an optimum around 50 CpGs per cell type and performed poorer as more CpGs were included. The *EMeth* algorithms appeared to perform relatively well for low numbers of marker CpGs ($n = 2$–10 per cell type). Finally, the *EMeth-Laplace* method was top-performing, both when using only a few marker CpGs ($n = 2$–10 per cell type), or when many markers were included CpGs ($n = 300$–500). It should be noted that these results simply highlight the robustness of algorithms to varying numbers of markers. The optimal number of loci may also be influenced by marker specificity.

## Deconvolving small fractions

These analyses capture a broad spectrum of cell compositions, with each blood cell type at a concentration between 0 and 75%. Biologically, these ranges are however more constrained, with neutrophils being more abundant than B or T-cells. To assess whether performance estimates also translate to biologically relevant cell abundances, we next evaluated deconvolution on 42 in silico mixtures, with proportions not generated randomly but selected from deconvolution estimates of bulk whole-blood samples ($n = 42$). This analysis yielded somewhat lower deconvolution performances, with lower $R^2$ due to more limited ranges of proportions (Supplementary Fig. 7a, b), and higher RMSE values for the more abundant cell types (Supplementary Fig. 7b, c). Nevertheless, the relative performance of the deconvolution methods was highly consistent with deconvolution on univariately generated proportions (Supplementary Fig. 7a).

These analyses indicate that deconvolution performance can be different for less abundant cell types. In many instances, such rare cell types are evident or of particular interest. In order to specifically test how predictions differed for these smaller fractions, we also reassessed performance exclusively for cell-type contributions below 3% on the original in silico dataset ($n = 200$). Accuracy at this threshold was noticeably lower compared to larger fractions (Fig. 6a). Interestingly, adding reference CpGs, from 2 to 100 CpGs, improved $R^2$ for small fractions (difference in $R^2 = 0.27$, $P = 1.079 \times 10^{-7}$, CI 95%: −0.36, −0.16) was observed, indicating that addition of reference CpGs is particularly beneficial for predicting small fractions, but also that small fractions are difficult to predict accurately, irrespective of the algorithm used (Figs. 6b and 5a). In conclusion, deconvolution for small fractions is inadequate in performance for all methods tested, but this can be mitigated to some extent by enlarging the reference marker panel.

## Impact of incomplete or over-extensive references

In many cases, the reference used for deconvolution can be inaccurate, by including more or fewer cell types than those present in a bulk sample. This can introduce noise into the deconvolution experiment.

Therefore, we compared deconvolution performances when one cell type was lacking from the reference, or when a cell type was included in the reference but absent from the in silico mixture. Removing cell types from the reference generally tends to improve deconvolution accuracy, except when a cell type is left out that is similar to another one included in the reference (e.g., CD4+ and CD8+ cell types; Supplementary Fig. 8a). Furthermore, regularization-based deconvolution methods such as elastic net and lasso, as well as *FARDEEP* deconvolution, performed very well in this experiment, significantly outperforming *Meth atlas* and *DCQ* (difference in RMSE = −0.04, $P = 6.507 \times 10^{-7}$, CI 95%: −0.05, −0.02; Supplementary Fig. 8b). On the other hand, including more cell types in the reference has a similar effect, but seemingly less intense, with slightly worse deconvolution accuracy for cell types that are highly similar (Supplementary Fig. 8c). In this setting, *ICeDT, EMeth-Laplace* and *Meth atlas* produced the best deconvolution results, significantly outperforming *DCQ, EMeth-Normal, EMeth-Binomial* and ridge deconvolution (difference in RMSE = −0.012, $P = 1.457 \times 10^{-9}$, CI 95%: −0.015, −0.010; Supplementary Fig. 8d). In general, an over-inclusive reference performs better than an incomplete one. Cell types absent from the in silico mixture were erroneously assigned fractions up to 18.3%, with the largest fraction found for CD8+ T-cells by DCQ, and the lowest fraction, −8%, found for CD8+ T-cells by *MethylResolver* (Supplementary Fig. 8e).

## Deconvolution of DNA methylation sequencing data

DNA methylation is increasingly being profiled by sequencing-based methods such as whole-genome, reduced representation, targeted or amplicon bisulfite sequencing (BS-seq), or third-generation nanopore-based sequencers. Here, DNA methylation levels are quantified by calculating the fraction of reads with a methylated CpG over all reads at a given locus, yielding data similar to array-based measurements. These profiles however differ from array-based profiles as they are count-based, quantifying the exact number of sequencing reads showing CpG methylation, rather than a percentage-based estimate of the fraction of methylated CpGs. Also, the selection of marker CpGs differs, with a much larger search space: all 28 million CpGs in the human genome can putatively serve for the selection of marker CpGs from whole-genome bisulfite sequencing (WGBS) data, versus only ~450,000 or ~850,000 CpGs available on microarrays. Additionally, instead of parsing individual CpGs, average DNA methylation over entire genomic regions can be leveraged for deconvolution, with DNA methylation at flanking CpGs being often highly correlated[44].

Here, we tested performance for sequencing-based DNA methylation data of the same deconvolution methods we describe above (Table 1). We performed deconvolution using non-overlapping genomic regions flanking 100 bp as markers. Including multiple flanking CpG reduces measurement errors and improve overall deconvolution performance (difference in $R^2 = 0.05$, $P < 10^{-16}$, CI 95%: 0.04, 0.06; Fig. 7 and Supplementary Fig. 9). Selected differentially methylated regions (DMRs) where relatively specific, both in datasets used for markers

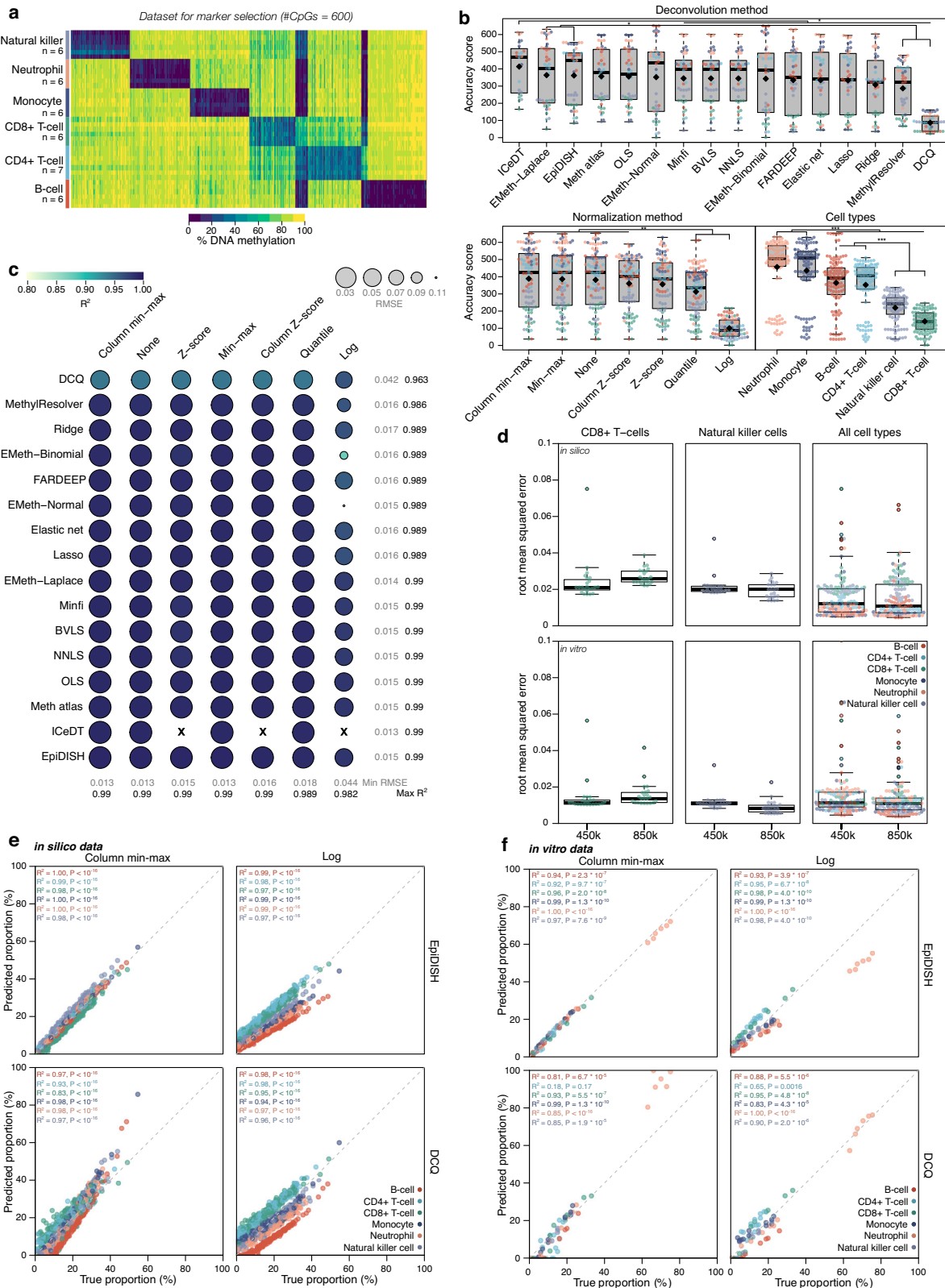

selection and mixture generation, however absolute methylation levels at marker loci differed considerably between datasets (Fig. 7a and Supplementary Fig. 10a–c). Deconvolution was performed on 100 in silico mixtures, comprising six immune cell types: neutrophils (fraction range: 0.86%–36.06%), monocytes (fraction range: 1.8%–54.7%), CD4+ T-cells (fraction range: 0.1%–36.3%), CD4+ T-cells (fraction range: 0.04%–35.9%), natural killer cells (fraction range:

0.1%–39.4%) and B-cells (fraction range: 0.3%–46.2%) (Supplementary Fig. 10d). CD8+ T-cell fractions were predicted less accurately than other cell types (difference in AS = −225.3, $P < 10^{-16}$, CI 95%: −254.3, −196.3; Fig. 7b).

Relative performance between deconvolution methods was highly comparable to array data for most algorithms, except for *MethylResolver* and *ICeDT* which were significantly outperformed by

**Fig. 4 | Deconvolution of immune cell types on Illumina EPIC array. a** Matrix of marker CpGs ($n = 600$) used for building immune cell methylation reference of EPIC data. Samples for six cell types were included: neutrophil ($n = 6$), natural killer cell ($n = 6$), B-cell ($n = 6$), CD4+ T-cell ($n = 7$), CD8+ T-cell ($n = 6$) and monocyte ($n = 6$). **b** Deconvolution accuracy represented as boxplots showing accuracy scores for the different deconvolution methods, normalization methods, and cell types on 200 in silico mixtures. Black diamond shapes represent median values, and colors represent cell types. The boxplots present median values and quartiles, whiskers the minimum and maximum values, and dots the individual data points. *P*-values were determined using two-tailed FDR-adjusted Dunn's tests. \**P* < 0.05, \*\**P* < 0.01, \*\*\**P* < 0.001. **c** Performance of deconvolution on 200 in silico mixtures for all algorithm-normalization combinations, represented as circles. Spearman's $R^2$ is represented by color, root mean squared error (RMSE) is represented by size. Rows show deconvolution algorithms, columns show normalization methods.

**d** Boxplots showing RMSE values of in silico ($n = 200$) and in vitro ($n = 12$) datasets for CD8+ T-cells, natural killer cells, and all cell types on CpGs selected for 450 K and EPIC array data. The boxplots present median values and quartiles, whiskers the minimum and maximum values, and dots the individual data points. **e** Scatter plots of 200 in silico mixtures showing true proportions (x-axis) and predicted proportions (y-axis) in percentages for the best-performing (left-upper) and worst-performing (right-lower) algorithm-normalization combinations on in silico EPIC data. $R^2$ and *p*-values were calculated using Spearman's rank correlation test. **f** Scatter plots of in vitro data ($n = 12$) showing true proportions (x-axis) and predicted proportions (y-axis) in percentages for the best-performing (left-upper) and worst-performing (right-lower) algorithm-normalization combinations on in silico EPIC data. $R^2$ and *p*-values were calculated using Spearman's rank correlation test. 'X' symbol represents missing values. Source data are provided as a Source Data file. Exact *p*-values are added in the Source Data file.

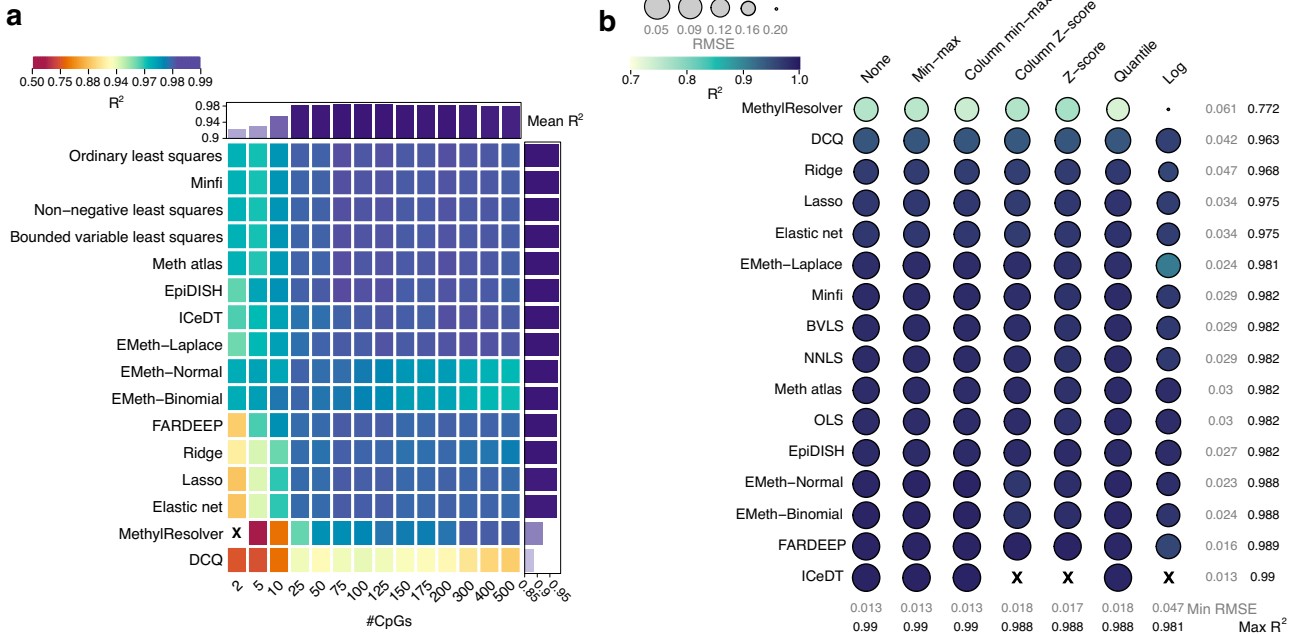

**Fig. 5 | Impact of reference size on deconvolution accuracy of immune cell types using EPIC array data. a** Heatmap showing Spearman's $R^2$ values on 200 in silico mixtures for all deconvolution methods at variable numbers of marker loci. **b** Performance of deconvolution on 200 in silico mixtures using a 10 CpG reference. Algorithm-normalization combinations are visualized as circles, with size representing root mean squared error and color representing Spearman's $R^2$. Rows show deconvolution algorithms, columns show normalization methods. 'X' symbol represents missing values. Source data are provided as a Source Data file.

most other algorithms (difference in AS = −109.7, $P = 1.03 \times 10^{-7}$, CI 95%: −151.3, −69.7). Normalization did not significantly impact the overall deconvolution performance (Fig. 7b–d), with unnormalized data producing top-performing results. Furthermore, we noted a lower overall deconvolution performance for WGBS, at 34 × sequencing depth, than for EPIC array data (min RMSE = 0.03 vs 0.01, max $R^2$ = 0.98 vs 0.99). This may be due to experimental differences, as WGBS protocols are often far less standardized between research groups, but an alternative explanation may be the difference in approach taken to generate in silico mixtures. Indeed, reads were mixed for WGBS, whereas DNA methylation levels were calculated by proportional weight-summing for array-derived data.

### Sequencing depth and evenness, and number of markers

Deconvolution experiments are often based on targeted BS-seq[45,46]. In these, both the number of marker regions assays and the sequencing depth significantly impact the analysis cost. We set out to test these variables, first by varying the number of marker regions for deconvolving six immune cell types on the same 100 in silico mixtures. This revealed that a local optimum was reached when 100 to 200 regions

per cell type were included for deconvolution (median $R^2$ = 0.98), irrespective of the deconvolution method used (Fig. 8a). We next assessed the impact of sequencing depth. We simulated depth ranges between 34× and 0.5× by downsampling, assessing the performance of the *EpiDISH* deconvolution algorithm on 100 unnormalized in silico mixtures and varying the number of marker regions between 2 and 500 per cell type (Fig. 8b). Interestingly when using over five marker regions, all simulated average sequencing depths exceeding 14-fold appeared to yield a similar performance, suggesting that a depth of ~14× suffices for deconvolution, and that accuracy is boosted more by including more marker regions than by higher sequencing depths. Lastly, also evenness of coverage may impact deconvolution, by altering the error rate of DNA methylation estimates. We therefore assessed the effect of coverage skewness by comparing the performance of deconvolution on mixtures with a regular 14 × coverage (mean 13.7, IQR 7–19) to mixture with the same average but an artificially less even coverage (mean 13.8, IQR 3–23). Such a more skewed coverage clearly affects performance when a small number of markers is used, and we observed only minor performance penalties when a large number of marker loci was deployed (>150 marker loci; Fig. 8c).

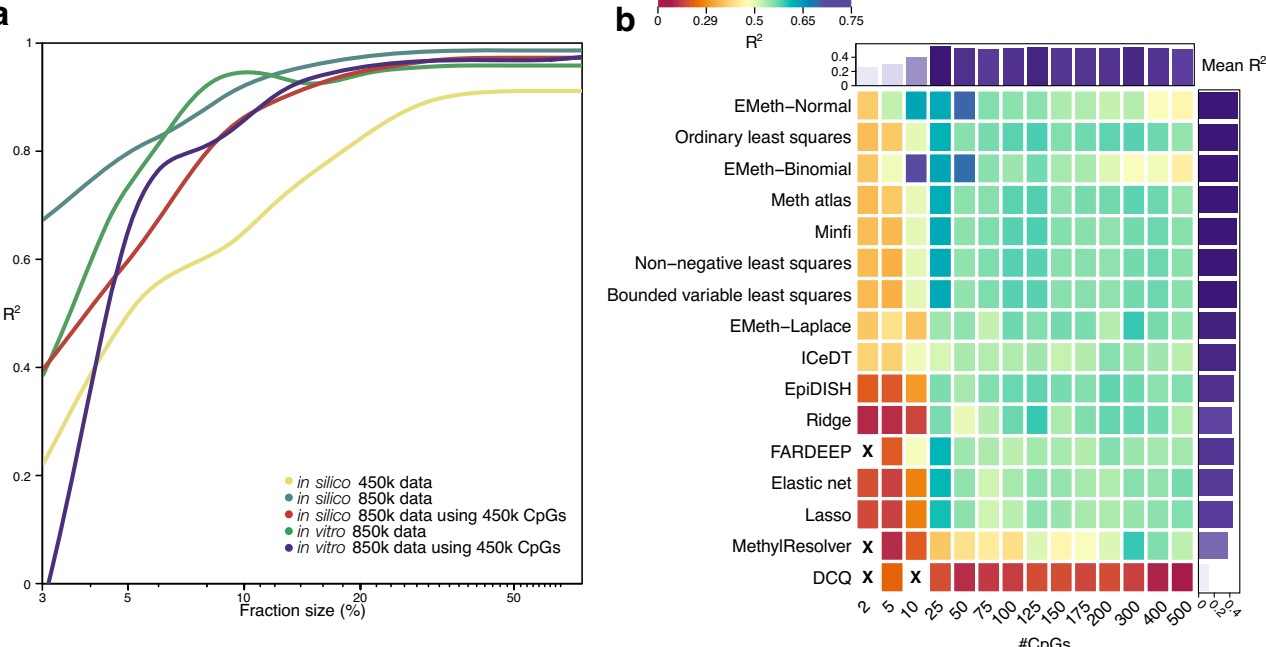

**Fig. 6 | Impact of reference size on deconvolution accuracy for small proportions of immune cell types using EPIC array data. a** Line plot showing cubic smoothed splines of Spearman's $R^2$ values (y-axis) for in silico and in vitro datasets at increasing fraction sizes, ranging from 3 to 75% (x-axis, log-scaled). **b** Heatmap showing Spearman's $R^2$ values for small fractions (<30%) on 200 in silico mixtures for all deconvolution methods at variable numbers of marker loci. 'X' symbol represents missing values. Source data are provided as a Source Data file.

## Combined analysis of deconvolution performance

Having modeled differences in various parameters, we finally compared the overall deconvolution accuracy across all datasets, by normalizing metrics per dataset using Z-score. *DCQ*, *EMeth-Normal* and *EMeth-Binomial* performed significantly worse than the other algorithms (difference in AS [Z-score] = −1.22, $P = 1.632 \times 10^{-5}$, CI 95%: −1.51, −0.64; Fig. 9a). Additionally, *EpiDISH* performs significantly better than most algorithms (difference in AS [Z-score] = 1.22, $P = 2.55 \times 10^{-4}$, CI 95%: 0.66, 1.84). Relative to the other algorithms, *Meth atlas* performed remarkably better on small fractions than on all fractions (Fig. 9b).

## Discussion

Despite the importance of cell fraction deconvolution for basic science and clinical studies, a large benchmark of deconvolution methods and DNA methylation analysis methods is lacking. A few studies have been performed, but these focused on a limited number of algorithms, without assessing the impact of cell fraction size, of normalization methods, of the types of input material, of the number of marker CpGs used as reference, and for sequencing-based methylome, of the impact of sequencing depth and evenness of coverage[26,27]. Here, we benchmarked 16 reference-based deconvolution methods and seven normalization methods, adding up to 109 combinations, both for deconvolution at the tissue and the cell-type level, and for array- and sequencing-based data. These were tested on in silico, in vitro and cytometry-quantified mixtures, enabling a quantitative head-to-head comparison between predicted and actual fractions. To facilitate future benchmarking of novel deconvolution algorithms, we provide an extensive Supplementary Data section containing metrics for the performed experiments with and without normalization, as well as all marker CpGs identified per cell/tissue type. Lastly, some metrics, such as root mean squared error and Jensen–Shannon divergence, might be biased towards either highly or lowly abundant cell types, depending on the research question, this should be taken into consideration. In relation to this, we refer to Supplementary Data 4–7 where we provide all performance metrics,

including Spearman's $R^2$, root mean squared error, Jensen–Shannon divergence, and accuracy score values.

Given the ubiquity of data normalization methods available, we anticipated data normalization to have a positive impact on deconvolution performance. Remarkably, this did not hold true for most algorithms, which mostly performed optimally without additional normalization. Another aspect influencing deconvolution performance is specificity of the marker loci. Illumina 450 K arrays interrogate over 450,000 CpG sites, and a further 413,743 CpGs are represented on the more recent EPIC arrays. This implies that only 1.7% or 3.2%, for 450 K and EPIC respectively, of all CpGs in the genome are considered for selection of tissue- or cell-specific CpGs. Therefore, fewer specific marker CpGs for any given cell type are available for selection. While this is not an issue when disparate cell types are deconvolved, highly discriminatory marker CpGs are scarcer for more similar cell types, such as natural killer and CD8+ T-cells. Another deconvolution variable is the number of marker loci. We observe that, for deconvolution of six cell types, increasing this number from 5 to 100 results in increasingly accurate predictions, while further increasing beyond 100 marker CpGs mostly yields only marginal gains for the tested algorithms. This may be due to overfitting at the marker selection step, and it should be noted that this optimum may differ when deconvoluting a variable number of cell types, or cell types of varying similarity. Also, the completeness of the reference should be considered, as both an over-extensive and incomplete reference can negatively affect deconvolution performance. Indeed, we observed a decrease in predictive accuracy when the number of cell types present in the reference differs from that in the mixture. However, if the cell types are unknown, a more comprehensive reference did perform better. As always, in an ideal scenario, references are used that accurately reflect the target population for optimal deconvolution. Finally, for DNA methylome sequencing data, the depth of sequencing will also drastically affect the deconvolution performance, while plateauing at 14× coverage using marker regions of 200 bp length. It should be noted however that for accurate prediction of excessively small

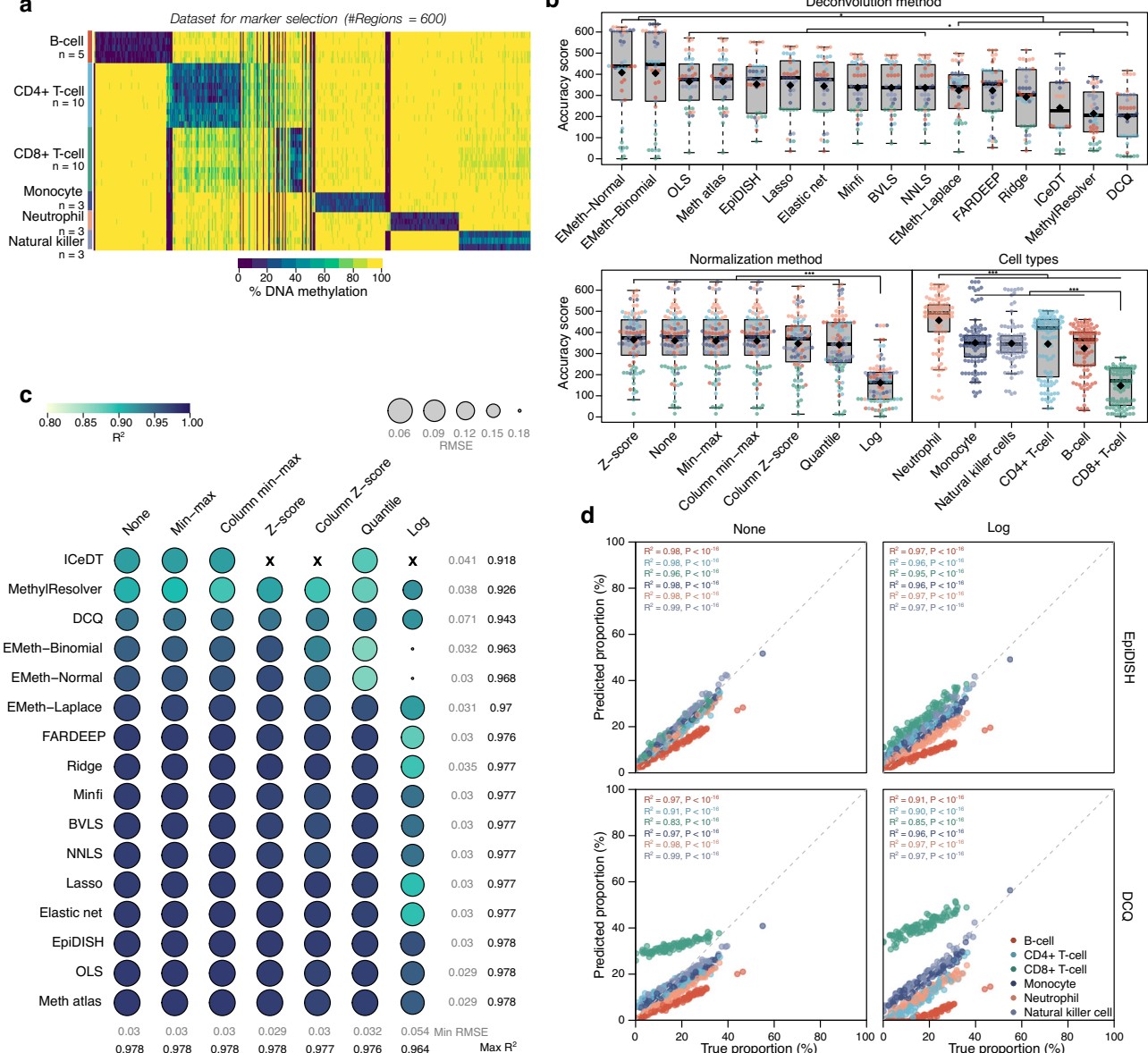

**Fig. 7 | Deconvolution of immune cell types from WGBS data. a** Matrix of marker regions (*n* = 600) used for building immune cell methylation reference of WGBS data. Samples for six cell types were included: neutrophil (*n* = 3), natural killer cell (*n* = 3), B-cell (*n* = 5), CD4+ T-cell (*n* = 10), CD8+ T-cell (*n* = 10), and monocyte (*n* = 3). **b** Boxplots showing accuracy scores for 16 different deconvolution methods, seven normalization methods, and six cell types on 100 in silico mixtures. Black diamond shapes: median values, colors: cell types. The boxplots present median values and quartiles, whiskers the minimum and maximum values, and dots the individual data points. *P*-values were determined using two-tailed FDR-adjusted Dunn's tests.

*$P < 0.05$, **$P < 0.01$, ***$P < 0.001$. **c** Performance of deconvolution on 100 in silico mixtures. Algorithm-normalization combinations are visualized as circles. Spearman's $R^2$ is represented by color, root mean squared error is represented by size. Rows show deconvolution algorithms, columns show normalization methods. **d** Scatter plots showing true (x-axis) and predicted proportions (y-axis) for the best (left-upper) and worst-performing (right-lower) algorithm-normalization combinations on 100 in silico mixtures. $R^2$ and *p*-values were calculated using Spearman's rank correlation test. 'X' symbol represents missing values. Source data are provided as a Source Data file. Exact *p*-values are added in the Source Data file.

fractions (i.e., <3%), sufficient sequencing depth and reference size is essential.

Differences between deconvolution methods are mostly attributable to the underlying statistical algorithms. The overall best-performing algorithm, *EpiDISH*, leverages a robust partial correlation (RPC) model. Its superior performance may be explained by its relative insensitivity to outlier values while still being able to pick up consistent signals coming from rare cell types. In contrast, *DCQ* was the worst-performing algorithm overall. It applies a variation on elastic net regression. Although this model is effective in preventing overfitting to the reference by reducing the number of features, it likely masks the more subtle signals coming from rare cell types. These rare cell types

are particularly well deconvolved by non-regularized linear regression models, such as ordinary least squares and *Meth atlas* (leveraging a non-negative least squares model). This might be due to non-negative least squares typically not allowing assignment of negative values to very small proportions, which would result in reduction of these proportions to 0% as seen in other algorithms.

Our study also has some limitations. Firstly, we focus on reference-based methodologies for benchmarking. Alternative algorithms have emerged over the past decade, but are more complex for benchmarking, and they also have different fields of application[47–49]. With the advent of DNA methylome maps for most major cell types in the human body, we anticipate that most future studies in patients will

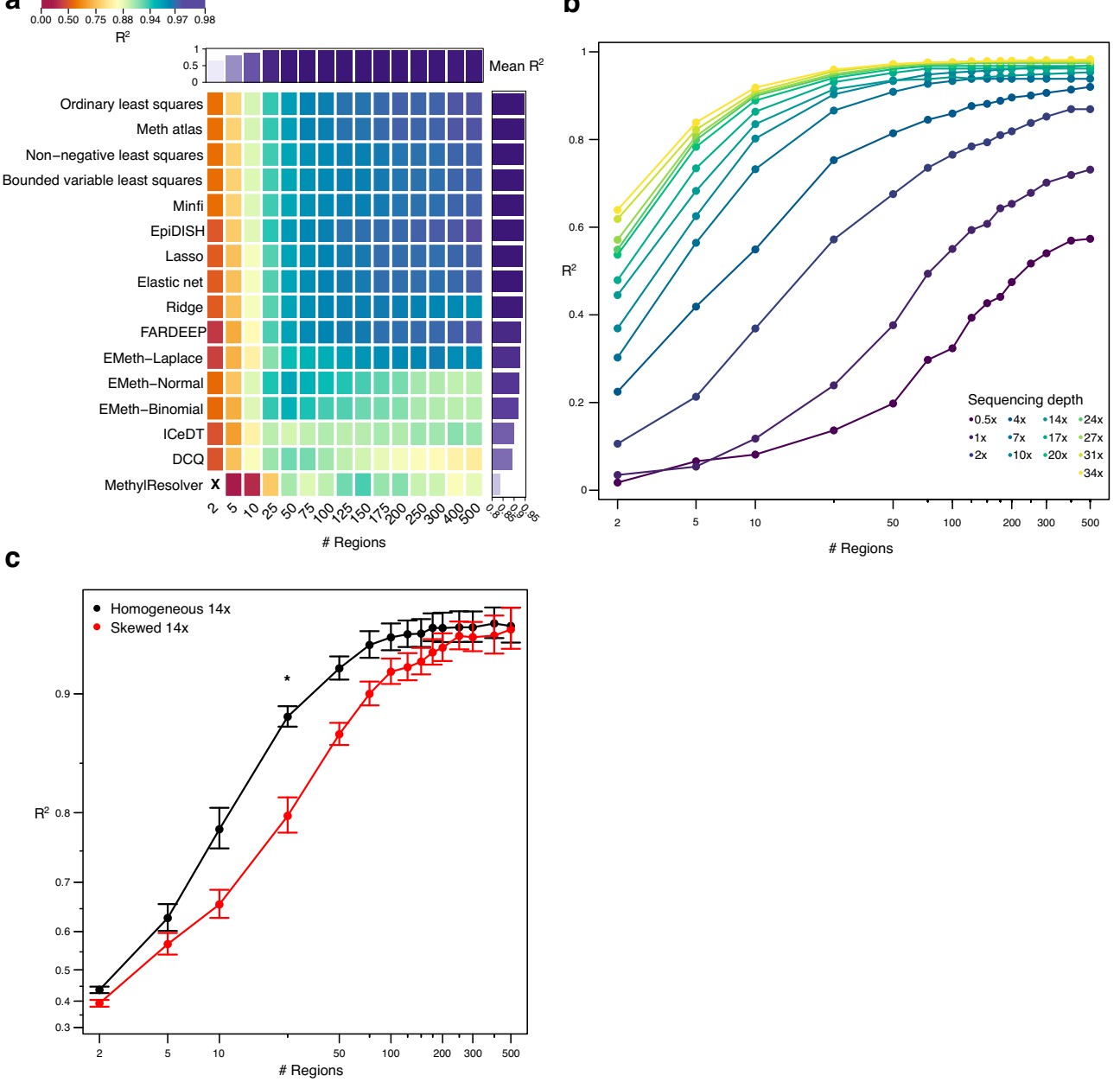

**Fig. 8 | Factors influencing sequencing-based deconvolution accuracy.**
**a** Heatmap showing Spearman's $R^2$ values on 100 in silico mixtures for all deconvolution methods at variable numbers of marker regions. **b** Line and scatter plot showing Spearman's $R^2$ values (y-axis) on 100 in silico mixtures for a variable number of marker regions per cell type (x-axis, log-scaled) at different sequencing depths. **c** Line and scatter plot showing Spearman's $R^2$ values (y-axis) on 100 in silico mixtures for a variable number of marker regions (x-axis, log-scaled) per cell type (x-axis, log-scaled) for skewed 13.8 (Q1 = 3, Q3 = 23) and homogeneous 13.7 (Q1 = 7, Q3 = 19) 14x covered sequencing data. Data are represented as mean ± SEM. *P*-values were calculated using a linear mixed model with deconvolution methods as random effects. *$P < 0.05$. 'X' symbol represents missing values. Source data are provided as a Source Data file.

rely on reference-based deconvolution[32]. Next, because of the limited availability of high-quality datasets, we selected marker loci from the datasets that were used to build reference matrices, which might induce overfitting. Additionally, read-based deconvolution has also been proposed to analyze sequencing data[50,51]. This has not been investigated here. Secondly, most benchmarking datasets we use are derived from in silico mixtures sampled from univariate uniform distributions. These may differ from real-life data but offer the advantage of being customizable in high throughput and representing exact ground truths. Indeed, real-life datasets typically lack accurately determined cell-type contributions, and they thus fail to serve as accurate benchmarks. Though sampling from a uniform distribution

allows direct comparison between cell types while controlling for typical differences in proportion, researchers may prefer to assess model performances using distributions reflecting real biological composition estimates, as exemplified in Fig. 3f and Supplementary Fig. 7. Furthermore, preprocessing methods for array data could have an impact on deconvolution performance and should be considered. As only data from healthy tissues is evaluated, deconvolution performance on samples from pathological specimens is unknown. For example, cancer is associated with pervasive perturbations of the DNA methylation landscape that is not reflected in the currently available reference profiles. Lastly, ease of implementation differs between deconvolution methods. For example, untailored methods (e.g., ridge,

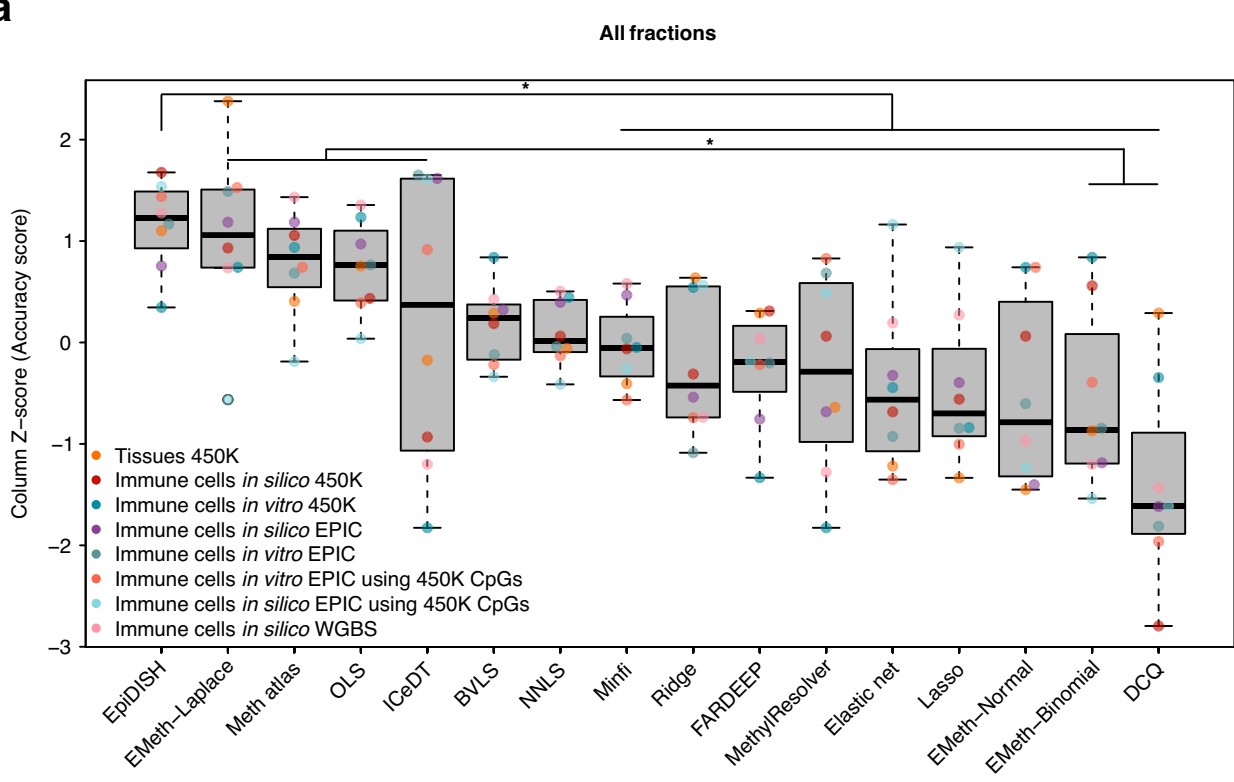

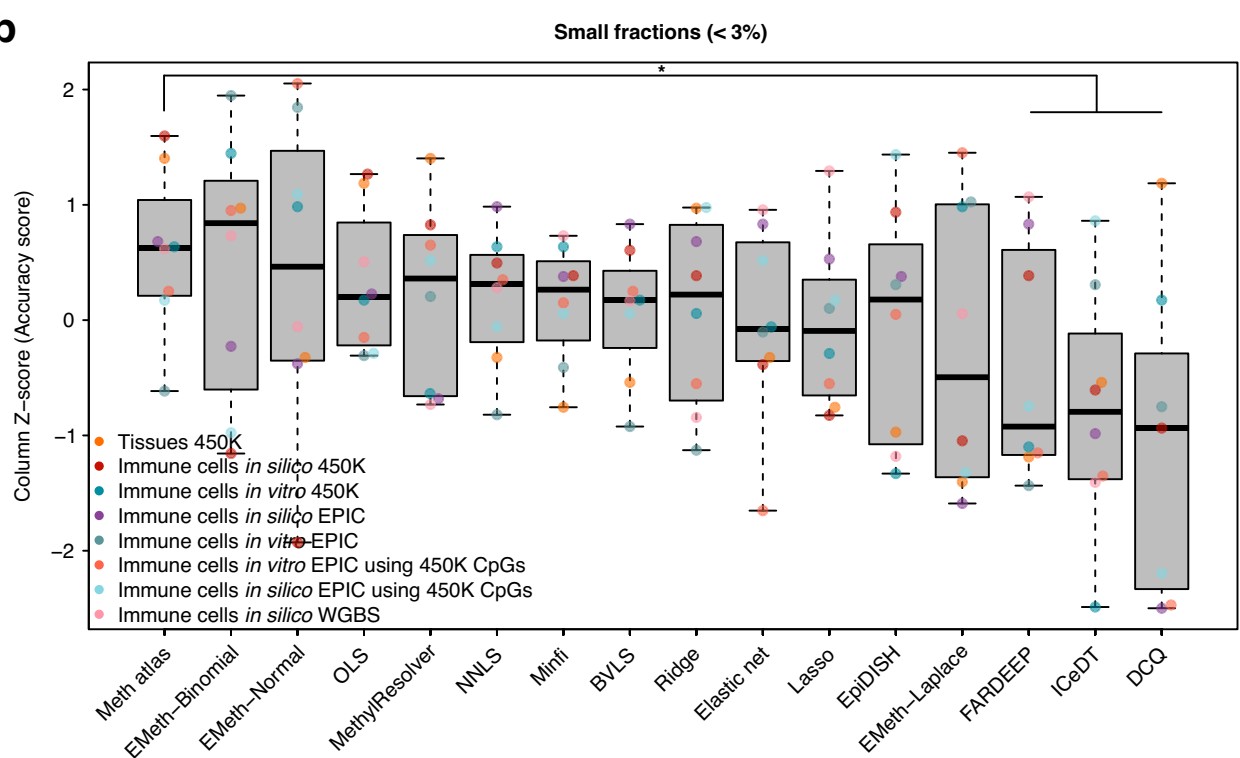

**Fig. 9 | Comparison of deconvolution accuracy across datasets. a**, **b** Boxplots showing accuracy scores, column Z-score normalized per dataset ($n = 8$ independent in silico and in vitro datasets), for each of the deconvolution methods in **a** all fractions and **b** fractions below 3%. The boxplots present median values and quartiles, whiskers the minimum and maximum values, and dots the individual data points. *P*-values were determined using two-tailed FDR-adjusted Dunn's tests. **P* < 0.05, ***P* < 0.01, ****P* < 0.001. Source data are provided as a Source Data file. Exact *p*-values are added in the Source Data file.

elastic net, and lasso regression) may be more difficult to implement, which should be considered for inexperienced users.

In conclusion, cell fraction deconvolution is a welcome alternative to expensive and time-consuming cell sorting techniques, but dependent on optimal algorithm and parameter selection. We provide a comprehensive benchmarking of most currently available reference-based DNA methylation deconvolution methods, comparing performances between different data formats and resolutions. Overall, we observe that normalization rarely positively impacts deconvolution and that *EpiDISH* consistently performs well in most contexts[31]. Furthermore, we provide guidelines on the appropriate number of loci that should ideally be used for deconvolution, and on the optimal sequencing depth needed to determine cell-type contributions to bulk samples analyzed with BS-seq data.

## Methods

No ethics evaluation was required for this study.

### Dataset selection

**Healthy tissues.** Illumina Infinium 450 K datasets for healthy tissues were identified on ArrayExpress. For each tissue, two datasets were used: one to perform marker selection and deconvolution, and one to generate in silico mixtures. Healthy tissues included kidney (marker selection: GSE59157; mixture generation: GSE50874), liver (marker selection: GSE61258; mixture generation: GSE61278), small intestine (marker selection: GSE73832; mixture generation: GSE50475) and blood (marker selection: GSE48472; mixture generation: GSE84003). In silico mixtures of all tissues were generated, which were subsequently used for assessment of deconvolution performance.

For generation of the in vitro datasets, DNA was extracted from purified natural killer cells, B-cells, monocytes, neutrophils, CD4+, and CD8+ T-cells, which were mixed in prespecified proportions and analyzed on EPIC arrays[43]. Cells were isolated using immunomagnetic labeling. Neutrophils were isolated by separating leukocytes using HetaSep followed by density gradient separation and neutrophil negative selection. All other cell types were negatively isolated from untouched peripheral mononuclear cells by indirect immunomagnetic cell labeling (i.e., CD14, CD19, CD56, CD4T, and CD8T). Proportions of whole-blood samples were quantified using flow cytometry. Lastly, in silico mixtures were constructed by selecting randomly generated fractions for each cell type and computing methylation values for each CpG of which each cell type contributes proportionally to its assigned fraction. For each in silico mixture, methylation signals for each of the cell types were sampled from one randomly selected sample of the set of methylome profiles for that cell type, to reflect the actual technical variation in samples between mixtures.

**Immune cells.** Immune cell array datasets were selected based on the availability of methylation data for either purified cell types or whole-blood samples with matched cytometry-determined proportions. Both array-based, i.e., 450 K HumanMethylation (marker selection: GSE71244, GSE65097; mixture generation: GSE35069, in vitro and cytometry-quantified: GSE77797) and EPIC (marker selection: GSE110554; mixture generation: GSE103541 and GSE129376; in vitro: GSE110554), and sequencing datasets were included so comparisons between data formats were possible.

For the marker selection of sequencing data, the GSE186458 dataset was used. Benchmarking of cell fraction deconvolution was performed on in silico sequencing read mixtures of several independent sequencing datasets acquired from the European Genome-Phenome archive: EGAD00001000710, EGAD00001001189, EGAD00001001261, EGAD00001001473, EGAD00001002460 and EGAD00001002508. For building these in silico mixtures, as total sequencing depth varied greatly between samples, all samples of one specific cell type were combined into 1 large mixture from which random sequencing reads were then sampled.

Proportions of array-based whole-blood samples were quantified using flow cytometry[40]. In silico mixtures were constructed by selecting randomly generated fractions for each cell type and computing methylation values for each CpG of which each cell type contributes proportionally to its assigned fraction. For each in silico mixture, methylation signals for each of the cell types were sampled from one randomly selected sample of the set of methylome profiles for that cell type, to reflect the actual technical variation in samples between mixtures. Alternatively, sequencing-based in silico mixtures were constructed similarly by selecting randomly generated fractions for each cell type and combining *n* sequencing reads of each cell type contributing proportionally to their specific assigned fraction.

### Marker selection

Reference CpGs were selected using an adaptation of the algorithm used by Luo et al. [39]. We initially selected CpGs for each of the k tissues/cell types showing Benjamini–Hochberg adjusted significant Welch two-sample *t*-test *p*-values (significance level of 5%) computed between methylation values of the target group and all other groups. The highest of all pairwise computed *p*-values was selected, to ensure high cell/tissue specificity. Secondly, from these CpGs we selected 100 CpGs with the highest mean methylation differences between the target group and all other groups. This resulted in k × 100 CpGs that make up the complete marker set.

For selection of DMRs used in WGBS deconvolution, we applied the same strategy with the added constrained that regions were not allowed to overlap.

### Processing of WGBS data

Raw sequencing reads were aligned to the GRCh37 genome using bwa-meth v0.2.5, trimmed using Trim Galore v0.6.6, deduplicated using Picard v3.1.0 and lastly methylation values were extracted using MethylDackel v0.6.0[52–55].

### Normalization workflow

Predictive performance of cell fractions was assessed between seven normalization conditions, including (column-wise) Z-score, (column-wise) min-max normalization, quantile normalization, log normalization, and no normalization.

Regular and column-wise Z-score normalization was performed by applying the following formula to the dataset:

$$f(x) = \frac{x - \mu}{\sigma} \tag{1}$$

Regular and column-wise min-max normalization was performed by applying the following formula to the dataset:

$$f(x) = \frac{x - Min(x)}{Max(x) - Min(x)} \tag{2}$$

Quantile normalization attempts to equalize two distributions in a rank-based way. For this study, we applied the 'normalize.quantiles' function of the 'preprocessCore' R-package v1.62.1[56].

Finally, log normalization was performed by applying the following formula to the dataset:

$$f(x) = log_e(x) \tag{3}$$

### Deconvolution workflow

A total of 16 deconvolution algorithms were included for comparisons. For linear models, cell fractions are reflected by the coefficients of the

model, as these coefficients resemble the contribution of each cell type. As the most basic model, we included ordinary least squares regression (OLS). Additionally, we also included several regularization models, such as elastic net, ridge, and lasso regression[29]. Some other constrained statistical algorithms included in the assay are bounded-variable least squares (BVLS), least trimmed squares (LTS), non-negative least squares (NNLS), linear constrained projection (CP), and robust partial correlation (RPC). For NNLS and BVLS, we applied the model without adjustments for deconvolution[28,35]. Additionally, we included several tailored deconvolution software packages, such as *Minfi* using CP, *MethylResolver* and *FARDEEP* using LTS, *DCQ* using elastic net regression, *Meth atlas* using NNLS, and *EpiDISH* using RPC[31–34,36,38]. Furthermore, expectation-maximization algorithms are leveraged by *EMeth-Binomial, EMeth-Laplace, EMeth-Normal,* and *ICeDT*[30,37].

Deconvolution algorithms were stripped from any inherent normalizations, such that only one normalization algorithm was applied at a time. Finally, the predictive accuracy of the algorithms between all the included conditions were compared.

## Statistics and reproducibility

Deconvolution accuracy was assessed using a set of 3 metrics: Spearman's $R^2$, Jensen–Shannon divergence (JSD) and root mean squared error (RMSE) between predicted values and actual values for cell fractions. Additionally, we evaluated an accuracy score, combining all 3 metrics:

$$Accuracy\ score = \frac{rank\left(R^2\right) + rank(RMSE) + rank(JSD)}{3} \quad (4)$$

Ranks were assigned based on the order of highest to lowest values for $R^2$ and lowest to highest values for RMSE and JSD. Normality was assessed using Shapiro–Wilk test. *P*-values were calculated by two-sided student's *t*-tests, Mann–Whitney *U*-tests, Dunn's tests, and Spearman's rank correlation. No statistical method was used to predetermine the sample size. Two samples were excluded from the analyses because of outlying methylation values. The experiments were not randomized. The Investigators were not blinded to allocation during experiments and outcome assessment.

## Software

All downstream analyses were performed using R v4.1.1 and Python v3.8.11.

## Reporting summary

Further information on research design is available in the Nature Portfolio Reporting Summary linked to this article.

## Data availability

The data used in this study from the Gene Expression Omnibus (GEO) are available under the following accession codes: GSE59157, GSE50874, GSE61258, GSE61278, GSE73832, GSE50475, GSE48472, GSE84003, GSE71244, GSE65097, GSE35069, GSE77797, GSE110554, GSE103541, GSE129376 and GSE186458. The data used in this study from the European Genome-phenome Archive (EGA) are available under the following accession codes after data access authorization: EGAD00001000710, EGAD00001001189, EGAD00001001261, EGAD00001001473, EGAD00001002460 and EGAD00001002508. Source data are provided with this paper.

## Code availability

Relevant code for simulation of deconvolution benchmarking on a demo dataset is available at https://github.com/FunctionalEpigeneticsLab/DNAme-deconvolution-benchmarking[57].

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

## Acknowledgements

We thank Alejandro Sifrim for assistance during the revision of the paper. Computing was performed at the Vlaams Supercomputer Center. This study was supported by Research Foundation – Flanders (FWO), grant nos. grant nos. S003422N, G0B4822N and G0C7519N (to B.T.); and by the Stichting tegen kanker, grant no F/2020/1544 (to B.T.). B.T. is supported by a BOFZAP mandate.

## Author contributions

B.T., K.D.R., and H.C. conceptualized and designed the study. B.T. and K.D.R. supervised the project. K.D.R. developed computational protocols and performed bioinformatics analyses. K.D.R., B.T., H.C., and K.L. contributed to the interpretation of results. B.T. and K.D.R. wrote the manuscript. All coauthors reviewed the manuscript.

## Competing interests

The authors declare no competing interests.
