## [Peer Review File · Nature Communications]

Benchmarking of Methods for DNA Methylation DeconvolutionReviewer #1 (Remarks to the Author):

De Ridder et al. benchmarked deconvolution methods for estimating cell-type proportions from DNA methylation. Deconvolution methods are central in standard pipelines of methylation analysis. Given that previous benchmarking studies were not very comprehensive and are now outdated due to the recent surge in proposed methods, a comprehensive evaluation of these methods can be valuable to the epigenetic community.

Most of the paper is clearly written, and the set of analyses is comprehensive. However, several major issues render the current conclusions unreliable. Specifically:

1. Evaluation metrics and procedures should be refined. R^2 should not, in general, be calculated across (non-normalized) proportions of different cell types. Consider Fig 3d, for example – strong linear concordance between estimated and measured neutrophil counts (which are easiest to estimate given that they constitute more than half of the composition in a typical whole-blood sample) can drive relatively high R^2 levels (i.e., across all cell types) regardless of the concordance between estimates and measured counts of other cell types. Put differently, if one were to permute all non-neutrophil estimates in Fig 3d, the R^2 would still be relatively high. Consequently, high R^2 levels can generally represent overly optimistic performance and obscure performance in lowly abundant cell types. As such, ranking the different methods based on the current R^2 evaluation is not reliable. There are several ways to address this, including calculating R^2 for each cell type separately (and then possibly taking the median or mean across cell types, at least for visualization purposes), normalizing the proportions and estimates of each cell type (separately) to be in the range $[0,1]$, or subtracting cell-type-level means (and possibly dividing by cell-type-level standard deviations) to project all compositions onto a comparable scale. The latter two are more reasonable for small datasets, for which separate evaluation of the different cell types is more limited.
2. The authors “randomly generated proportions” in their data simulation. No further details are provided, which makes it impossible to clearly judge the results of their analysis. Based on Figures S2c and S3d, it appears as if cell proportions were sampled from the same distributions and likely not sampled from a joint distribution but rather each cell type was sampled separately from a univariate distribution. In that case, the cell compositions in the simulations do not reflect real-world distributions of cell compositions. Importantly, the distribution used can have a great effect on performance. As the authors report, all deconvolution methods underperform in estimating lowly abundant cell types compared to highly abundant cell types. In addition, sampling proportions from univariate distributions are expected to result in weaker levels of correlations between the fractions of the different cell types compared to the correlation in real biological samples. There are several ways to address these issues. For example, the authors can apply deconvolution to large bulk methylation data and can then draw cell-type proportion profiles from this large pool of estimated cell-type compositions, which should provide a decent representation of real biological cell composition profiles. Alternatively, the authors can simulate from a distribution such as Dirichlet (which would need justified parameterization). Both approaches are expected to result in more realistic distributions of cell-type fractions.
3. It appears that in each of their experiments, the authors reported the deconvolution results of a single pseudo-bulk mixture. Reliably benchmarking the different methods must account for the uncertainty merely stemming from sampling. The authors should evaluate the methods across multiple simulated datasets in each experiment. While the number of methylomes available for simulating mixtures is clearly limited, the cell composition differences can be very large (see point 2 above). Different simulations can therefore correspond to different cell composition profiles used.
4. RMSE is often more emphasized in the results compared to R^2 (e.g., Figures 2-4). However, most downstream applications that utilize cell-type proportion estimates carry out linear analysis, thus justifying a focus on metrics of linear concordance, such as R^2 . In addition, it would be informative to set an RMSE baseline performance – for each experiment, it would be informative to know what the RMSE of a naïve method that estimates the same cell-type proportions profile for all samples with the cell-type population mean as the estimate of each cell type. This baseline level will help to highlight in which experiments the RMSE metric is more informative.
5. “To confidently apply deconvolution, a method is expected to be robust to technical variables such as varying wet-lab protocols, data sources, and data formats.” – Another potentially critical source of inconsistent variation across studies is inter-individual variation that may affect many CpGs (e.g., age, sex, or ancestry). Specifically, reference data representing a certain population

may not be proper for the deconvolution of a different population (see, for example, Yousefi et al. 2015). The authors should discuss to what extent (if at all) their evaluations provide insight into this potential issue. Can the authors design an analysis to address this question? (e.g., evaluate the performance of deconvolution under different levels of similarity between the reference and the deconvolved bulk profiles).

6. The authors report deconvolution results as a function of the number of CpGs used. Since the reference data is also being used for feature selection, there is a risk of overfitting and including non-informative CpGs in the deconvolution panel. While this appears to not be a concern when using a relatively small number of CpGs (possibly due to having DMRs with very strong effects), it may become a concern when including larger numbers of CpGs. In other words, one possible explanation for the decrease in performance may be the result of overfitting. One way to address this concern would be to have at least one analysis in which features are not selected directly from the reference profiles but rather from independent reference samples. If such an analysis is not possible for some reason, then the authors should justify why and acknowledge this limitation.

Minor comments:

- The number of bulk profiles used is not clearly reported in every simulation.
- The authors did not specify which statistical test was used in their marker selection procedure.
- I believe the correct term is "deconvolve," not "deconvolute."
- Legends identifying cell types are missing in many figures (particularly in scatter plots).
- The authors report results under the metric "RMSE [column Z-score normalized]." It is not clear to me what this metric is and why the reported values are negative.

Reviewer #2 (Remarks to the Author):

The authors comprehensively evaluated 13 deconvolution algorithms (and 91 algorithm+normalization combinations) of DNA methylation-based cell-type deconvolution algorithms. The authors concluded the influence of sequencing depth on deconvolution performance and the complexity of how performance depends on different variables. Overall, I find the work shed useful insight to the DNA methylation research community. For example, the authors discussed how the deconvolution accuracy depends on reference, cell fraction and normalization methods. However, I think the work could be improved by including some mathematical intuition of why certain inaccurate results arise. More methods and some discussion of the context with other recent comparison work of the same topic should also be included to make the work more complete. Below are my detailed suggestions:

For maximal transparency, authors should publicly post their analysis script as Rmarkdown and Jupyter Notebook.

Many notable methods in this domain are not included. For example, Houseman 2012, SparsePCA, ReFACTor, IDOL, GLINT, CONFINED, Tsisal, CAMDAC, EpiSCORE, FARDEEP, RPC, DCQ, ICeDT, scMD, ...

Are the normalization methods compared actually used in prior studies? The authors listed 7 data normalization methods and went to length to discuss their impact and combination with algorithms. But no reference is provided to suggest their usage in prior studies. I personally never saw them used in practice or in the literature. Throughout the paper, the author reported no added benefit of data normalization. I am not surprised to see this result as many of the listed normalizations, e.g., quantile normalization, Z-score normalization, log transform, obviously introduce the global change of the methylation level reading. Certain normalization seems more often used in omics data processing (e.g., Log transformation of count-based signals). I suggest the authors either provide some background or simplify the analysis by removing benchmarking the normalization methods or moving them to the supplemental material. I find the discussion of algorithm-normalization pair distracting (e.g., in Fig 3).

Related to the comment above, DNA methylation array data is indeed normalized by having the

background subtracted and dye bias mitigated. How would these more relevant "normalizations" such as background subtraction, dye bias correction, BMIQ, dash, noob, pOOBAH, etc., affect deconvolution accuracy?

The observations the authors report are very technical and descriptive. I suggest the authors distill some mathematical/mechanistic insight to the finding. For example, why do rare cell types tend to be poorly deconvolved? How to evaluate an RMSE difference of 0.01? Should one worry about this for biological interpretation? Could the authors explain why certain algorithms perform better?

Figure 2 uses bulk tissue methylation as the base component of mixture. The bulk tissue itself can be heterogeneous. The authors should only consider doing this experiment using purified cell type data. I find this figure of limited interest to most readers.

Figure 2b, the mean methylation ratio difference is not a good indicator for specificity as it could be driven by one or a few outlier cell types. Consider maybe an F test. Why did the validation dataset have higher marker CG methylation specificity for small intestine?

Most analyses in this work use the author's own method of reference construction. This is based on one-vs-rest pairwise comparison with mean methylation difference. More advanced methods have been developed in the field (e.g., PMID 37525279, or IDOL). Since the reference construction is very critical, it's worth covering how these deconvolution methods perform using alternative strategies. How does the reference marker selection impact deconvolution accuracy?

In practice, not every cell type has a similar number of markers (as limited by either the assay platform or biology). In such a case, what would be the best marker selection strategy for cell type deconvolution?

Fig S3b-c, it is not clear what reference and validation datasets are. Is the poor performance of natural killer and CD8+ T cell because of the suboptimal reference quality or the biological characteristics of these cells, as discussed in on line 142?

The optimal number of marker CpGs would depend on the accuracy/precision of methylation measurement of each CpG and the quality of marker CpG selection.

Another variable associated with sequencing data is uniformity. 24x uniform coverage is different from 24x highly skewed coverage. How does this affect deconvolution performance?

In the discussion section, reference-free algorithms and read-based deconvolution methods have been mentioned but without providing references. Please provide.

In multiple places, the authors used the term "higher" to refer "text above". I got confused for a second and would suggest alternative wording to improve readability.

Reviewer #3 (Remarks to the Author):

In this manuscript, the authors compared the performance of different algorithms in deconvoluting cell type fractions from bulk DNA methylation data. First of all, the authors have done a lot of work. However, they did not properly cite earlier work. The comparison itself has some flaws and is not comprehensive. In addition, the overall conclusion seems not to be very informative. Thus I feel that the paper in its current form is not suitable for publication. Comments detailed below:

1. The authors did not cite previous work that performed comparison on a similar/same task. Jeong et al. 2022 Briefings in Bioinformatics. "Systematic evaluation of cell-type deconvolution pipelines for sequencing-based bulk DNA methylomes". Song et al, 2022, Briefings in Bioinformatics. "A systematic assessment of cell type deconvolution algorithms for DNA methylation data". These papers should at least be cited and discussed. And the authors should highlight the novelty of their work in light of these previous benchmark studies.

2. The authors compared 13 algorithms, many of which are not tailored for DNA methylation data, but rather simple regression models, such as LASSO and Ridge regression. And their performances are very similar. I am not sure how informative it is to include these general regression models in comparison. For a typical user, they would not be able to implement these regression models and use them in their own data analysis. Also, I recommend the authors list these algorithms in the main text (can break down into several categories), and briefly introduce their core algorithms.

3. The authors decided to use $n=100$ markers per cell for deconvolution. It is not justified in the paper. From Figure 5a, it looks like the performance of most algorithms further increases when more than 100 CpGs are included. Will including more markers per cell significantly increase the run time of the algorithms? Furthermore, the authors performed marker selection using their own-developed algorithms. Is this algorithm a common practice? Marker selection may have a significant impact on the performance, however, the authors have not evaluated different ways of selecting markers.

4. It is not clear to me what in vitro and in vivo datasets are in the paper. Could the authors explain them in the methods? In line 181 it says "To further validate these in silico analyses, we next assessed performances on DNA extracted from different cell types, mixed in vitro at prespecified ratios¹⁴", however ref 14 is a review paper.

5. Line 176, it says "array probes described higher". Should be "...described earlier"? This occurred multiple times throughout the manuscript.

We are grateful to each of the 3 reviewers for the positive and very constructive comments they made on our work, allowing us to further improve the quality of the study, and to the editor for agreeing to review a new version of our study including extra analyses and deconvolution methods proposed by the reviewers. We have made substantial efforts to address each of the comments, as detailed below. Key additions include sections in which we more fully assess the method of marker selection, add new algorithms, perform analyses assessing technical variability, validate performance on real-life proportions and sequencing depth skewness, as well as more in-depth mechanistic interpretation of the results.

REVIEWER COMMENTS

Reviewer #1 (Remarks to the Author):

De Ridder et al. benchmarked deconvolution methods for estimating cell-type proportions from DNA methylation. Deconvolution methods are central in standard pipelines of methylation analysis. Given that previous benchmarking studies were not very comprehensive and are now outdated due to the recent surge in proposed methods, a comprehensive evaluation of these methods can be valuable to the epigenetic community. Most of the paper is clearly written, and the set of analyses is comprehensive. However, several major issues render the current conclusions unreliable. Specifically:

1. Evaluation metrics and procedures should be refined. R^2 should not, in general, be calculated across (non-normalized) proportions of different cell types. Consider Fig 3d, for example – strong linear concordance between estimated and measured neutrophil counts (which are easiest to estimate given that they constitute more than half of the composition in a typical whole-blood sample) can drive relatively high R^2 levels (i.e., across all cell types) regardless of the concordance between estimates and measured counts of other cell types. Put differently, if one were to permute all non-neutrophil estimates in Fig 3d, the R^2 would still be relatively high. Consequently, high R^2 levels can generally represent overly optimistic performance and obscure performance in lowly abundant cell types. As such, ranking the different methods based on the current R^2 evaluation is not reliable. There are several ways to address this, including calculating R^2 for each cell type separately (and then possibly taking the median or mean across cell types, at least for visualization purposes), normalizing the proportions and estimates of each cell type (separately) to be in the range [0,1], or subtracting cell-type-level means (and possibly dividing by cell-type-level standard deviations) to project all compositions onto a comparable scale. The latter two are more reasonable for small datasets, for which separate evaluation of the different cell types is more limited.

We thank the reviewer for this insightful remark. We acknowledge the evaluation metrics could be further refined, and have addressed this in 3 different ways:

1. Computation of R^2 is indeed ideally performed separately **for each cell type**, as suggested by the reviewer, and we have now implemented this throughout the manuscript, calculating R^2 per cell type and reporting their average per algorithm/normalization combination.
2. To further refine the evaluation, we now add the **Jensen-Shannon divergence (JSD)**, a performance metric assessing homogeneity between predicted and actual fraction distributions. This is a third metric, in addition to the RMSE and the R^2 .
3. We expanded performance scoring by assessing an additional summary metric, representing a combination of R^2 , RMSE and JSD ranks. We call this the “**accuracy score**”, which provides a more general characterization of deconvolution performance.

$$Accuracy\ score = \frac{rank(R^2) + rank(RMSE) + rank(JSD)}{3}$$

2. The authors “randomly generated proportions” in their data simulation. No further details are provided, which makes it impossible to clearly judge the results of their analysis. Based on Figures S2c and S3d, it appears as if cell proportions were sampled from the same distributions and likely not sampled from a joint distribution but rather each cell type was sampled separately from a univariate distribution. In that case, the cell compositions in the simulations do not reflect real-world distributions of cell compositions. Importantly, the distribution used can have a great effect on performance. As the authors report, all deconvolution methods underperform in estimating lowly abundant cell types compared to highly abundant cell types. In addition, sampling proportions from univariate distributions are expected to result in weaker levels of correlations between the fractions of the different cell types compared to the correlation in real biological samples. There are several ways to address these issues. For example, the authors can apply deconvolution to large bulk methylation data and can then draw cell-type proportion profiles from this large pool of estimated cell-type compositions, which should provide a decent representation of real biological cell composition profiles. Alternatively, the authors can simulate from a distribution such as Dirichlet (which would need justified parameterization). Both approaches are expected to result in more realistic distributions of cell-type fractions.

We thank the reviewer for noting our lack of detail in how proportions were generated. The distributions we selected indeed may not reflect real-world biology, but were randomly generated proportions, sampled from a univariate distribution. We now describe this explicitly in the manuscript (Line 99-101). The underlying rationale is 3-fold:

1. Samples with a known mixture composition are very rare: determining mixture composition of real-life data is mostly dependent on dissociation which typically skews towards some cell types and fails to recover others, and/or on assessing marker expression by immunostaining, which is limited by the availability of markers, antibodies and other technical limitations. Mixtures with known compositions are hence rare.
2. Estimating mixture compositions by deconvoluting bulk data will yield reasonable estimates but stand to overestimate performances, as they typically require modeling using the same data as our test data. Indeed, as described, some cell types show poorly concordant methylation profiles and are hence prone to overfitting.
3. Along the same lines, limiting the estimates of cell type contributions is more likely to yield erroneous performances: if a major cell type has an exceptionally poor reference, it will affect the performance of all other cell types, yielding inaccurate model performances.

We nevertheless acknowledge the merit of evaluating real-life distributions, especially when faced with a specific research question. We therefore retained our initial analyses, but now include an analysis of deconvolution on *in silico* mixtures for which proportions were directly taken from deconvoluted proportions of 42 independent whole-blood samples (GSE141682). Importantly, these yield a relative ranking of methods that is similar to the ranking obtained from sampling random distributions (rebuttal figure 1), thus further validating the original approach. These data were added to the manuscript (Line 272-284)

Notably, in addition to this, we also present results of deconvolution on FACS-quantified proportions from 5 blood samples and 12 *in vitro* mixed samples (GSE77797). These also reflect real-world proportions. Importantly, they again yield a relative ranking of methods that is similar to that obtained from sampling from univariate distributions. Finally, concordance with real-world biology is also dependent on performance in small fractions (<3%), which is presented in Figure 6 and on lines 201-207 of the revised manuscript.

In conclusion, the observed concordances in performance between random and real-life distributions further validate our approach. We however acknowledge the issue and added to the discussion:

“Though sampling from a univariate distribution allows direct comparison between cell types while controlling for typical differences in proportion, researchers may prefer to assess model

performances using distributions reflecting real biological composition estimates, as exemplified in Figures 3f and S7.”

Rebuttal figure 1: a) Circle heatmap showing performance metrics for all algorithm-normalization combinations on *in silico* mixtures generated from real-life proportions (size represents RMSE, color represents R²). b) Boxplots showing R² (top) and RMSE (bottom) values for each cell type in *in silico* mixtures with proportions generated from a univariate distribution and real-life proportions. ‘X’ symbol represents missing values.

3. It appears that in each of their experiments, the authors reported the deconvolution results of a single pseudo-bulk mixture. Reliably benchmarking the different methods must account for the uncertainty merely stemming from sampling. The authors should evaluate the methods across multiple simulated datasets in each experiment. While the number of methylomes available for simulating mixtures is clearly limited, the cell composition differences can be very large (see point 2 above). Different simulations can therefore correspond to different cell composition profiles used.

We thank the reviewer for this comment. For each experiment, we provide summary statistics and 95% confidence intervals of deconvolution results of 200 *in silico* mixtures, overall resulting in homogeneous distributions of proportions for each of the different cell types, allowing assessment of performance over the entire spectrum between 0-50% fractions. Additionally, for each *in silico* mixture, methylation signals for each of the cell types were sampled from one randomly selected sample of the set of methylome profiles for that cell type, so as to reflect the actual technical variation in samples between mixtures. The latter aspect is now described in more detail in the main manuscript (Line 103-105) and in the methods section (Line 515-517). In addition, we also assessed performances on 5 cytometry-quantified and 12 *in vitro* mixed 450k blood samples (GSE77797) and on 12 *in vitro* mixed EPIC blood samples (GSE110554) (Line 201-207 and 234-238). As such, we assume this to be sufficient to obtain a reliable and comprehensive estimate of both theoretical and biological deconvolution performances.

4. RMSE is often more emphasized in the results compared to R^2 (e.g., Figures 2-4). However, most downstream applications that utilize cell-type proportion estimates carry out linear analysis, thus justifying a focus on metrics of linear concordance, such as R^2 . In addition, it would be informative to set an RMSE baseline performance – for each experiment, it would be informative to know what the RMSE of a naïve method that estimates the same cell-type proportions profile for all samples with the cell-type population mean as the estimate of each cell type. This baseline level will help to highlight in which experiments the RMSE metric is more informative.

We thank the reviewer for this remark. We agree that overall R^2 is more informative for most applications. Therefore, as the reviewer suggested, focus was shifted more towards R^2 , as well as to our new summary metric, combining ranks of R^2 , RMSE and JSD (see Q1), for more comprehensive performance assessment.

5. “To confidently apply deconvolution, a method is expected to be robust to technical variables such as varying wet-lab protocols, data sources, and data formats.” – Another potentially critical source of inconsistent variation across studies is inter-individual variation that may affect many CpGs (e.g., age, sex, or ancestry). Specifically, reference data representing a certain population may not be proper for the deconvolution of a different population (see, for example, Yousefi et al. 2015). The authors should discuss to what extent (if at all) their evaluations provide insight into this potential issue. Can the authors design an analysis to address this question? (e.g., evaluate the performance of deconvolution under different levels of similarity between the reference and the deconvolved bulk profiles).

We thank the reviewer for highlighting this potential source of variation. To address this comment, we now include an analysis where samples from a single dataset are randomly assigned either to the reference or validation cohort. With this approach, systematic technical and population-specific variation is absent, while sex- and age-associated variation remains (age-range: 19-49 in reference vs 20-59 in validation cohort; sex-distribution (female/male): 5/13 in reference vs 1/17 in validation cohort). As expected, this resulted in better deconvolution performance overall, with median R^2 increasing from 0.991 to 0.996 ($P < 10^{-16}$; rebuttal figure 2). Importantly however, the different methods showed a similar ranking ($R = 0.69$), indicating that this source of variation does not affect the choice of the optimal method. These data were added to the manuscript (Line 240-251)

Rebuttal figure 2: a) Circle heatmap showing performance metrics for all algorithm-normalization combinations on *in silico* mixtures generated from dataset without technical variation (size represents RMSE, color represents R^2). b) Boxplots showing R^2 (top) and RMSE (bottom) values for each cell type in *in silico* mixtures generated from datasets with and without technical variation. 'X' symbol represents missing values. c) Scatter plot showing accuracy scores for all algorithm-normalization combinations from datasets with (x-axis) and without (y-axis) technical variability.

6. The authors report deconvolution results as a function of the number of CpGs used. Since the reference data is also being used for feature selection, there is a risk of overfitting and including non-informative CpGs in the deconvolution panel. While this appears to not be a concern when using a relatively small number of CpGs (possibly due to having DMRs with very strong effects), it may become a concern when including larger numbers of CpGs. In other words, one possible explanation for the decrease in performance may be the result of overfitting. One way to address this concern would be to have at least one analysis in which features are not selected directly from the reference profiles but rather form independent

reference sampels. If such an analysis is not possible for some reason, then the authors should justify why and acknowledge this limitation.

We thank the reviewer for this insightful remark and acknowledge that overfitting might indeed pose a potential problem when using a larger number of marker loci. Though it might indeed be of interest to look further into why performance drops when too many marker loci are leveraged, the paucity in datasets available prevents us from addressing this confounder. We have now acknowledged this limitation in the manuscript (Line 415-417, 431-433, 442-444).

Minor comments:

- The number of bulk profiles used in not clearly reported in every simulation.
This data was added to all figure legends.
- The authors did not specify which statistical test was used in their marker selection procedure.
This data was added to the materials and methods section (Line 523-530).
- I believe the correct term is "deconvolve," not "deconvolute."
Dictionaries fail to agree with each other on the matter. We have adjusted all instances to "deconvolve", as suggested.
- Legends identifying cell types are missing in many figures (particularly in scatter plots).
This information was added to all figure legends.
- The authors report results under the metric "RMSE [column Z-score normalized]." It is not clear to me what this metric is and why the reported values are negative.
These comments have now been resolved in the revised manuscript. These metrics were normalized within each dataset, using Z-score, such that comparison between datasets was possible. Therefore, values can be below 0.

Reviewer #2 (Remarks to the Author):

The authors comprehensively evaluated 13 deconvolution algorithms (and 91 algorithm+normalization combinations) of DNA methylation-based cell-type deconvolution algorithms. The authors concluded the influence of sequencing depth on deconvolution performance and the complexity of how performance depends on different variables. Overall, I find the work shed useful insight to the DNA methylation research community. For example, the authors discussed how the deconvolution accuracy depends on reference, cell fraction and normalization methods. However, I think the work could be improved by including some mathematical intuition of why certain inaccurate results arise. More methods and some discussion of the context with other recent comparison work of the same topic should also be included to make the work more complete. Below are my detailed suggestions:

For maximal transparency, authors should publicly post their analysis script as Rmarkdown and Jupyter Notebook.

We thank the reviewer for this suggestion. We now provide a Jupyter notebook for testing the deconvolution benchmark. This is described in the code availability statement (Line 599-601).

Many notable methods in this domain are not included. For example, Houseman 2012, SparsePCA, ReFACTor, IDOL, GLINT, CONFINED, Tsisal, CAMDAC, EpiSCORE, FARDEEP, RPC, DCQ, ICeDT, scMD, ...

We thank the reviewer for this comment, and for pointing us to 14 potential additional methods to test. This was addressed as follows:

We already included 3 of these methods in our study before the revision:

- *EpiSCORE* utilizes EpiDISH, and uses *RPC* for deconvolution.
- The Houseman algorithm was named *Minfi* and included as such.

We indeed overlooked 3 methods (FARDEEP, DCQ and ICeDT) and have now included these in our revised manuscript.

Some of the suggested methods do not return cell proportions as output. These 4 are SparsePCA, ReFACTor, IDOL and GLINT. They were not included.

Finally, 3 of the suggested methods are either reference-free and thus explicitly excluded from analysis (Tsisal and CAMDAC), or not designed for bulk data (scMD).

Are the normalization methods compared actually used in prior studies? The authors listed 7 data normalization methods and went to length to discuss their impact and combination with algorithms. But no reference is provided to suggest their usage in prior studies. I personally never saw them used in practice or in the literature. Throughout the paper, the author reported no added benefit of data normalization. I am not surprised to see this result as many of the listed normalizations, e.g., quantile normalization, Z-score normalization, log transform, obviously introduce the global change of the methylation level reading. Certain normalization seems more often used in omics data processing (e.g., Log transformation of count-based signals). I suggest the authors either provide some background or simplify the analysis by removing benchmarking the normalization methods or moving them to the supplemental material. I find the discussion of algorithm-normalization pair distracting (e.g., in Fig 3).

We thank the reviewer for this comment. Data normalization is a step that should always be considered when inferring information from a test dataset by use of an independently generated dataset showing technical and biological variability. Indeed, several deconvolution methods rely on data normalization, proposing different methods as a default (e.g., *Minfi* and *FARDEEP* applying quantile normalization). This renders their direct comparison unfeasible.

We hence decided to disregard the normalizations associated to each deconvolution method, and to separately normalize the data using different approaches that are commonly used in DNA methylation-based or transcriptome-based deconvolution (Cobos et al., Nature Communications 2020). Although most methods appear relatively insensitive to normalization in our study, evaluation of these methods is of interest to researchers and we therefore prefer to tackle this issue openly. Furthermore, even though benefits of normalization are indeed often minor, some specific algorithm-normalization combinations seem to outperform non-normalized deconvolution (Figure 2c). Nevertheless, we acknowledge that testing for these different normalizations complicates the figures and analyses, and have hence moved part of them to the supplementary figures.

Related to the comment above, DNA methylation array data is indeed normalized by having the background subtracted and dye bias mitigated. How would these more relevant “normalizations” such as background subtraction, dye bias correction, BMIQ, dash, noob, pOOBAH, etc., affect deconvolution accuracy?

We thank the reviewer for this insightful remark. Indeed, these array preprocessing methods may influence deconvolution performance. We however opted not to pursue this line of investigation for the following reasons: 1. These preprocessing steps are well established and typically applied in a standard analysis. 2. We have no access to raw IDAT files for most array datasets, which is a prerequisite for such analyses. 3. These normalizations are not applicable for sequencing data. We however added this as an aspect to consider in the discussion (Line 454-455).

The observations the authors report are very technical and descriptive. I suggest the authors distill some mathematical/mechanistic insight to the finding. For example, why do rare cell types tend to be poorly deconvolved? How to evaluate an RMSE difference of 0.01? Should one worry about this for biological interpretation? Could the authors explain why certain algorithms perform better?

We thank the reviewer for bringing this to our attention. Rare cell types are naturally more difficult to deconvolve as their contribution to the signal is extremely small and therefore more difficult to separate from manyfold larger proportions. Deconvolution performance on these small fractions greatly differs between algorithms (Figure 6b). These differences are obviously attributable to the underlying statistical methods that are leveraged. The overall best performing method, *EpiDISH*, works based on a robust partial correlation (RPC) model. The relative insensitivity to outlier values while still being able to pick up consistent signals coming from rare cell types might explain its improved performance compared to most of the other algorithms. In contrast, DCQ applies a variation on elastic net regression. Though this model is effective in preventing overfitting to the reference by reducing the number of features, this might mask the more subtle signals coming from rare cell types. For non-regularized regression models however, such as ordinary least squares and *Meth atlas* (leveraging a non-negative least squares model) we observe the best performance on rare cell types. This insight was described in the discussion section (Line 382-383, 433-435).

An RMSE difference of 0.01 in this case could be interpreted as a deviation of, on average, 1% between predicted fractions and actual fractions. This clarification was added to the revised manuscript (Line 116-118). Evaluation of this metric between different fraction sizes is however difficult, and thus difficult to compare between datasets. Therefore, researchers are generally more interested in not absolute but relative differences between cell types, e.g., relatively higher NK cell proportions in cancer patients after immune checkpoint inhibitor treatment, indicating that R^2 might be a more informative metric to evaluate for biological interpretation of deconvolution

results. For each experiment and method we now provide R^2 , RMSE, JSD and accuracy score metrics in supplementary tables S5, S6, S7 and S8 respectively, allowing researchers to evaluate these based on their specific needs.

Figure 2 uses bulk tissue methylation as the base component of mixture. The bulk tissue itself can be heterogeneous. The authors should only consider doing this experiment using purified cell type data. I find this figure of limited interest to most readers.

We thank the reviewer for this comment. It is true that in some bulk deconvolution problems (e.g., tissue samples) this kind of deconvolution is uninformative. However, for analysis of e.g., blood-derived cell-free DNA (cfDNA), coming from several different tissues and cell types, this is highly informative and more accurate than deconvolving based on profiles of purified cell types. Such liquid-biopsy-based deconvolutions are rapidly gaining traction in epigenetic biomarker research. Moreover, given the difficulties of purifying cells from tissues, the latter are also only available for a limited set of cell types, and we could not identify instances where 2 independent sets of DNA methylomes from the same cellular source are available.

Figure 2b, the mean methylation ratio difference is not a good indicator for specificity as it could be driven by one or a few outlier cell types. Consider maybe an F test.

We thank the reviewer for this insightful comment. We agree that an F-test is more reliable for assessing cell marker specificity. This has now been added to the figures and manuscript throughout.

Why did the validation dataset have higher marker CG methylation specificity for small intestine?

Though some of the small intestine samples for the reference dataset seem to be quite noisy, our CpG marker selection algorithm did manage to identify specific marker loci. The validation dataset was apparently less noisy, resulting in higher specificity for these same marker loci (this is visible when comparing the small intestine-specific marker loci between both heatmaps in Figure 2a). We investigated clinical variables annotated to the reference dataset but could not identify variables explaining this low signal specificity. In order not to complicate the manuscript further, we opted not to discuss this in the manuscript.

Most analyses in this work use the author's own method of reference construction. This is based on one-vs-rest pairwise comparison with mean methylation difference. More advanced methods have been developed in the field (e.g., PMID 37525279, or IDOL). Since the reference construction is very critical, it's worth covering how these deconvolution methods perform using alternative strategies. How does the reference marker selection impact deconvolution accuracy?

We thank the reviewer for this important remark. Indeed, method of reference construction is crucial to successful deconvolution. The method currently leveraged throughout the study is the one used in PMID 37525279, with as a sole difference that we required p-values below an FDR of 0.05 instead of an unadjusted p value of 0.05. We selected this method of marker selection so that we could both identify markers with the largest effect sizes while simultaneously being able to ensure equal representation of all cell types in the reference. Furthermore, we also performed a comparison between deconvolution with a reference built using our method and using the IDOL method mentioned (43/400 CpGs overlap between selection methods). This shows an overall better performance, both in terms of R^2 and RMSE, of our one-vs-rest pairwise strategy (rebuttal figure 3). These data have now been added to the manuscript (Line 172-174).

Rebuttal figure 3: Boxplots showing R^2 (left) and RMSE (right) values for each cell type between deconvolutions generated using reference markers selected by IDOL and custom algorithm. P-values were computed using Wilcoxon rank-sum test. *** = $P < 0.0001$.

In practice, not every cell type has a similar number of markers (as limited by either the assay platform or biology). In such a case, what would be the best marker selection strategy for cell type deconvolution?

The reviewer is correct in noting that different cell types or assays will yield different numbers of markers, warranting a more catered marker selection strategy. While certainly an interesting question, we feel that it is beyond the scope of our current manuscript, which provides an already elaborate analysis on varying numbers of markers in both array and sequencing data as well as for rare cell types and varying sequencing depths. Nevertheless, our analysis showing the effect of the number of markers on deconvolution performance could serve as a basis for developing a strategy to identify the optimal number of markers for each cell type. This is added to the discussion (Line 412-417)

Fig S3b-c, it is not clear what reference and validation datasets are.

Thank you for noting this. We refer to 'reference' datasets as those datasets that were used for both marker selection and used as the reference matrix used during deconvolution, while 'validation' datasets are the datasets used to build the *in silico* mixtures. We apologize for this confusion and have now adjusted this in the revised manuscript.

Is the poor performance of natural killer and CD8+ T cell because of the suboptimal reference quality or the biological characteristics of these cells, as discussed in on line 142? The optimal number of marker CpGs would depend on the accuracy/precision of methylation measurement of each CpG and the quality of marker CpG selection.

We thank the reviewer for this insightful remark. We observe high intra-cell type correlation of methylation signal at marker loci within the different cell types ($R^2 = 0.9 - 1.0$), indicating that reference quality high (rebuttal figure 4). Furthermore, we observe higher correlation between CD4+ and CD8+ T cells ($R^2 = 0.1-0.2$) as well as CD8+ and natural killer cells ($R^2 = 0.0-0.1$) compared to other cell type pairs ($R^2 = 0.0$). This indicates that because of the shared biological characteristics between these cell types, markers are less specific and therefore deconvolution performance is lower.

Rebuttal figure 4: Heatmap showing R^2 values of methylation ratios between all samples in reference dataset.

Another variable associated with sequencing data is uniformity. 24x uniform coverage is different from 24x highly skewed coverage. How does this affect deconvolution performance? We thank the reviewer for the interesting question. To address this, we compared deconvolution of regular 14x covered mixtures (mean 13.7, $Q1 = 7$, $Q2 = 19$) to mixtures for which the marker loci had a similar but more variable average coverage of 14x (mean 13.8, $Q1 = 3$, $Q2 = 23$). Performance was significantly lower in the skewed dataset when using <100 reference markers, and only minor differences in performance were observed when applying deconvolution with more markers (rebuttal figure 5).

Rebuttal figure 5: Scatterplot showing R^2 values for homogeneous (black) and skewed (red) 14x covered data using variable number of marker regions (x-axis). Error bars showing standard error of the mean.

In the discussion section, reference-free algorithms and read-based deconvolution methods have been mentioned but without providing references. Please provide. These have now been added (Line 439-440).

In multiple places, the authors used the term “higher” to refer “text above”. I got confused for a second and would suggest alternative wording to improve readability. We thank the reviewer for bringing this to our attention. This has now been adjusted in the manuscript throughout.

Reviewer #3 (Remarks to the Author):

In this manuscript, the authors compared the performance of different algorithms in deconvoluting cell type fractions from bulk DNA methylation data. First of all, the authors have done a lot of work. However, they did not properly cite earlier work. The comparison itself has some flaws and is not comprehensive. In addition, the overall conclusion seems not to be very informative. Thus I feel that the paper in its current form is not suitable for publication.

We acknowledge that several limitations were still present in the manuscript, as the reviewer mentioned. These have now been resolved. We hope that the study is now deemed sufficiently comprehensive, as we include a large number of methods as well as in-depth evaluation using 3 different performance metrics of several variables that affect deconvolution performance, such as marker selection, number of reference markers, technical and population variability and proportion distributions. Lastly, we provide a more in-depth discussion of the overall conclusions, highlighting shortcomings and advantages of the different potential approaches that should be considered when designing a deconvolution experiment.

Comments detailed below:

1. The authors did not cite previous work that performed comparison on a similar/same task. Jeong et al. 2022 Briefings in Bioinformatics. "Systematic evaluation of cell-type deconvolution pipelines for sequencing-based bulk DNA methylomes". Song et al, 2022, Briefings in Bioinformatics. "A systematic assessment of cell type deconvolution algorithms for DNA methylation data". These papers should at least be cited and discussed. And the authors should highlight the novelty of their work in light of these previous benchmark studies.

We thank the reviewer for bringing this to our attention. We now cited and discussed these references in the revised manuscript, as well as highlighting the novelty of our work (Line 69-76).

The authors compared 13 algorithms, many of which are not tailored for DNA methylation data, but rather simple regression models, such as LASSO and Ridge regression. And their performances are very similar. I am not sure how informative it is to include these general regression models in comparison. For a typical user, they would not be able to implement these regression models and use them in their own data analysis.

We thank the reviewer for this comment. Note that we added 3 additional deconvolution algorithms to the benchmark, adding up to 16 algorithms, of which 7 are tailored for methylation data (*EMeth-Binomial*, *EMeth-Normal*, *EMeth-Laplace*, *EpiDISH*, *MethylResolver*, *Minfi* and *MethAtlas*). Because of the paucity of reference-based algorithms specifically tailored for methylation data and as the deconvolution problem is often similar, most included algorithms are either simple linear models or models that were initially designed for transcriptome data, which seem to generally perform well in this setting. Furthermore, as both us and Cobos et al. (Nature Communications 2020) demonstrate, simple regression models can produce relatively accurate deconvolution results and can serve as a valuable baseline reference to compare other algorithms. Therefore, we do believe this can be of interest to potential readers as well as more experienced users. To further accommodate this comment, we now also discuss the ease of implementation of algorithms as a factor to consider when selecting an algorithm (Line 454). Furthermore, we provide both scripts and a Jupyter notebook which could serve as a starting point for potential users (Line 459-461).

Also, I recommend the authors list these algorithms in the main text (can break down into several categories), and briefly introduce their core algorithms.

We thank the reviewer for this advice. The algorithms have now been listed in the first results paragraph of the revised manuscript (Line 81-92).

The authors decided to use $n=100$ markers per cell for deconvolution. It is not justified in the paper. From Figure 5a, it looks like the performance of most algorithms further increases when

more than 100 CpGs are included. Will including more markers per cell significantly increase the run time of the algorithms?

We thank the reviewer for this comment. Indeed, including more markers significantly increases runtime for several algorithms. Therefore, we selected a total of 100 marker loci per cell type as a local optimum, as no significant performance increase is observed on increasing number of markers further, which is the case for all algorithms. We apologize for not clarifying this in the previous version of this manuscript, this has now been explained in the revised manuscript (Line 260-263).

Furthermore, the authors performed marker selection using their own-developed algorithms. Is this algorithm a common practice? Marker selection may have a significant impact on the performance, however, the authors have not evaluated different ways of selecting markers.

We thank the reviewer for this remark. Variations of this algorithm are commonly used depending on the research question (e.g., Luo *et al.*, Genome Med. 2023). To address the reviewer's concern, we have now added an extra analysis in which we compare deconvolution performance using the current algorithm and the more regularly used IDOL algorithm (Koestler *et al.*, BMC Bioinformatics, 2016), showing an overall better performance, both in terms of R^2 and RMSE, of our one-vs-rest pairwise strategy (rebuttal figure 6). The added benefit of the currently applied algorithm is that we can force equal contribution of all cell types to the reference, enabling us to make a more objective comparison between cell types.

Rebuttal figure 6: Boxplots showing R^2 (left) and RMSE (right) values for each cell type between deconvolutions generated using reference markers selected by IDOL and custom algorithm. P-values were computed using Wilcoxon rank-sum test. *** = $P < 0.0001$.

It is not clear to me what in vitro and in vivo datasets are in the paper. Could the authors explain them in the methods?

In line 181 it says "To further validate these in silico analyses, we next assessed performances on DNA extracted from different cell types, mixed in vitro at prespecified ratios¹⁴", however ref 14 is a review paper.

We apologize for not clearly annotating which identifiers correspond to which publicly available datasets (*in vitro*: GSE110554, GSE77797; cytometry-quantified: GSE77797), this has now been annotated more clearly in the revised manuscript. These datasets were previously generated by other research teams.

For generation of the *in vitro* datasets, they extracted DNA from MACS-purified natural killer cells, B cells, monocytes, neutrophils, CD4+ and CD8+ T cells, which were mixed in prespecified proportions and analyzed on EPIC arrays. Cells were isolated using immunomagnetic labelling. Neutrophils were isolated by separating leukocytes using HetaSep followed by density gradient separation and neutrophil negative selection. All other cell types were negatively isolated from untouched peripheral mononuclear cells by indirect immunomagnetic cell labeling (i.e., CD14, CD19, CD56, CD4T and CD8T). Proportions of whole-blood samples were quantified using flow cytometry.

Line 176, it says "array probes described higher". Should be "...described earlier"? This occurred multiple times throughout the manuscript.

We thank the reviewer for bringing these errors to our attention. They have now been corrected throughout in the revised manuscript.

Reviewer #1 (Remarks to the Author):

The authors did not adequately address three of my major comments, see below. I have two additional general comments:

- The authors should break text into smaller paragraphs for readability where applicable.
- Poor figure resolution makes it impossible to read some of the results, text, and legends in nearly all figures (one notable example is Figure 3).

Major comment #2:

First, despite my comment, I still could not find any description in the manuscript of how cell-type fractions were sampled for the in-silico simulations. Were the fractions sampled from Uniform(0,1)?

The authors make three points to justify their simulation design of using a univariate distribution for sampling cell proportions (which I assume are coming from Uniform(0,1) in the absence of other information). I believe these points are not well justified:

1. The fact that known compositions are rare is not an excuse. This issue can be addressed, for example, using one of the two approaches I suggested – using an empirical distribution of estimated compositions obtained by applying deconvolution to large bulk data or drawing from a multivariate Dirichlet distribution (parameters can be fitted using the suggested empirical distribution); note that I did not suggest using real cell compositions because those are indeed rare for large populations. Surely, such cell composition distributions will reflect actual cell composition distributions better than what one could get by sampling from Uniform (0,1). The authors suggest that approximations of real cell composition distributions are expected to be skewed, however, it is hard to imagine anything that would skew real distributions more than a simple Uniform(0,1).
2. The authors claim that my second suggested approach would overestimate performance. This is not true: the authors should think of deconvolution-based estimates as a pool of cell compositions from which they can randomly draw a composition profile for a given simulated mixture. The pool of deconvolution-based estimates can be derived from a dataset unrelated to the analysis, and therefore the claim about overestimation does not hold.
3. The authors suggest poor reference for some cell types as a reason to prefer sampling from Uniform (0,1) over empirical or parametric sampling from a multivariate distribution. However, a benchmarking of deconvolution methods will only be helpful to the epigenetic community if it captures the complexities of real scenarios. In particular, if reference quality indeed affects the quality of deconvolution, then researchers should be informed of which methods are more robust to such quality issues. Biasing the evaluation to ignore the complexities of the data may bias the conclusions of the authors and their recommendations and therefore misrepresent the best methods.

In summary, real cell compositions demonstrate strong correlations between cell types. Such correlations are not captured by Uniform (0,1). Furthermore, as the authors reported, deconvolution methods underperform in estimating lowly abundant cell types. Therefore, conclusions from simulated mixtures based on compositions sampled from Uniform (0,1) are expected to be very limited in reliably ranking methods by their expected performance in real-world applications.

Comment #1:

The authors added an evaluation of the R^2 metric per cell type, as I suggested. The authors added the JSD and an accuracy score they defined based on performance ranks. JSD and especially the rank-based evaluations are great additions to the benchmarking, however, the issue of sensitivity to cell-type abundance remains. All three metrics, R^2 , JSD, and RMSE are sensitive to cell-type abundance: if one or two cell types are, on average, much more abundant than the other cell types in the tissue (e.g., neutrophils in whole-blood mixtures), then the values in all three metrics, for a given method, will be dominated by the estimates of the abundant cell types. In whole-blood samples, for example, such benchmarking is likely to highlight the methods that are best at estimating neutrophils and not necessarily the methods that have the best overall performance (i.e., across cell types). While my concern is not an issue when sampling cell compositions from Uniform (0,1) (since there is no difference in abundance between cell types in this case), this might lead to severe biases when considering more realistic cell compositions, in

which different cell types may drastically differ in their abundance.

Comment #5:

The authors' response is unrelated to my comment. I raised the question of whether reference data representing a specific population will be limited in the deconvolution of a different population. One example would be restricting the reference to include only aged individuals for deconvolving mixtures from young individuals (see my reference to Yousefi et al. 2015 as an example of why this question is of interest).

Reviewer #2 (Remarks to the Author):

The authors have adequately addressed my comments. Please consider the paper for publication.

Reviewer #3 (Remarks to the Author):

My concerns have been addressed.

Reviewer #3 (Remarks on code availability):

Only demo data benchmark is provided on Github.

We are grateful to both reviewer 2 and 3 for the positive feedback and for recommending publication, as well as to reviewer 1 for the additional comments to improve our manuscript. Below, we provide a rebuttal to the additional comments of reviewer 1.

REVIEWER COMMENTS

Reviewer #1 (Remarks to the Author):

The authors did not adequately address three of my major comments, see below. I have two additional general comments:

- The authors should break text into smaller paragraphs for readability where applicable.
We thank the reviewer for bringing this to our attention. We have now structured the text into 30 smaller paragraphs.
- Poor figure resolution makes it impossible to read some of the results, text, and legends in nearly all figures (one notable example is Figure 3).
We thank the reviewer for bringing this to our attention. This has been addressed in the updated manuscript. As compression of figures by the manuscript management system is beyond our control, we also upload raw uncompressed PDF files.

Major comment #2:

First, despite my comment, I still could not find any description in the manuscript of how cell-type fractions were sampled for the in-silico simulations. Were the fractions sampled from Uniform(0,1)?

In our previous revision, we added throughout the manuscript that cell fractions for in silico simulations were sampled from a univariate distribution, but we realize that this should have been defined explicitly. We apologize for this oversight and thank the reviewer for spotting this. We now added to the Manuscript: “*Individual fractions were sampled from a uniform univariate distribution ranging between 0 and 1, after which values were rescaled to add up to 1*” (Line 105-106).

The authors make three points to justify their simulation design of using a univariate distribution for sampling cell proportions (which I assume are coming from Uniform(0,1) in the absence of other information). I believe these points are not well justified:

Below, we provide a rebuttal to the criticism of the reviewer on our 3 points. We would however like to preempt this discussion with the following notions:

1. We designed our study to assess variables that influence deconvolution performance in an unbiased manner. Using uniform univariate distributions, it is straightforward to disentangle confounders related to similarity between cell types, cell type abundance and technical differences, but more difficult when limiting analyses to a restricted range of biologically relevant fractions.
2. Because we used a Uniform(0,1) sampling distribution, several *in silico* mixtures will be included that mimic the prior of Dirichlet-sampled biological distributions from whole blood, alongside a range of other scenarios not exactly aligning with the expected healthy state, making the assessment more generalizable.
3. In our manuscript, we do not propose to provide accuracy estimates for deconvolution algorithms. These are dependent on individual research questions and invariably a function of sample composition. We merely aim to compare the relative performance of deconvolution algorithms under different scenarios.
4. We already included, in the revised manuscript, 3 separate analyses on deconvolution performance in real-live datasets. Each of these analyses shows that, while uniform univariate distributions yield higher accuracy metrics, the relative ranking of deconvolution algorithms remains stable. This indicates that

relative performances can be reliably estimated from univariate distributions and demonstrates that the goal of our study (helping researchers to select the optimal deconvolution algorithm) can be attained using this strategy.

The fact that known compositions are rare is not an excuse. This issue can be addressed, for example, using one of the two approaches I suggested – using an empirical distribution of estimated compositions obtained by applying deconvolution to large bulk data or drawing from a multivariate Dirichlet distribution (parameters can be fitted using the suggested empirical distribution); note that I did not suggest using real cell compositions because those are indeed rare for large populations. Surely, such cell composition distributions will reflect actual cell composition distributions better than what one could get by sampling from Uniform (0,1). The authors suggest that approximations of real cell composition distributions are expected to be skewed, however, it is hard to imagine anything that would skew real distributions more than a simple Uniform(0,1).

We thank the reviewer for this comment. The comment we made that known compositions are rare obviously refers to “real cell compositions”.

As the reviewer can appreciate, we added an analysis using one of the approaches he/she suggested: we deconvolved *in silico* mixtures with proportions generated by deconvolving whole-blood samples. We acknowledge that these better reflect one biological situation. However, we observe rankings of algorithm-normalization combinations that are similar between uniform distributions and real-life distributions sampled from estimates based on deconvolution of a large dataset (rebuttal figure 1a), with the same algorithms raking top (*ICeDT*) and bottom (*DCQ*) s. This indicates that rankings are not significantly influenced by how proportions are generated. Given the arguments laid out higher and our choices in study design, we prefer to retain the analysis using uniform distributions in the manuscript.

Rebuttal figure 1: a) Scatterplot showing correlation between accuracy scores for algorithm-normalization combinations on *in silico* mixtures generated using real-life distributions (x-axis) and uniform distributions (y-axis). b) Ridgeline plot showing proportion densities for a uniform sampling distribution (left) and a real-life distribution (right).

2. The authors claim that my second suggested approach would overestimate performance. This is not true: the authors should think of deconvolution-based estimates as a pool of cell compositions from which they can randomly draw a composition profile for a given simulated mixture. The pool of deconvolution-based estimates can be derived from a dataset unrelated to the analysis, and therefore the claim about overestimation does not hold.

We thank the reviewer for this comment. We understand the reviewer's point. However, under the proposed scenario, the make-up of the pool of cell compositions is dependent on the deconvolution method. This make-up can be skewed to cater for shortcomings of specific algorithms (e.g. zero-inflation). For example, using a poorly performing algorithm such as lasso regression, deconvolution models that similarly depend on regression perform better, but not when using *EpiDISH* for generating proportions (rebuttal figure 2a). To demonstrate this explicitly, we compared the ranking of each deconvolution algorithm when that algorithm was used for generating pools of cell compositions, to the median rank of that algorithm when any other algorithm was used (rebuttal figure 2b). This showed that 10 out of 16 algorithms ranked higher when the pool of cells was generated using the same algorithm than when it was generated using a different algorithm, while 3 showed no altered rank and only 3 showed a lower rank. Overall, this confirms our statement that the rank of a deconvolution algorithm is improved when that algorithm is also used to estimate the pool of cell compositions.

Rebuttal figure 2: a) Boxplots showing accuracy score values for each deconvolution method on *in silico* mixtures with real-life proportions generated using *EpiDISH* and lasso regression. b) Scatterplot showing ranks based on median accuracy scores for proportions generated by the algorithm itself (x-axis) or the other algorithms (y-axis).

The authors suggest poor reference for some cell types as a reason to prefer sampling from Uniform (0,1) over empirical or parametric sampling from a multivariate distribution. However, a benchmarking of deconvolution methods will only be helpful to the epigenetic community if it captures the complexities of real scenarios. In particular, if reference quality indeed affects the quality of deconvolution, then researchers should be informed of which methods are more robust to such quality issues. Biasing the evaluation to ignore the complexities of the data may bias the conclusions of the authors and their recommendations and therefore misrepresent the best methods. In summary, real cell compositions demonstrate strong correlations between cell types. Such correlations are not captured by Uniform (0,1). Furthermore, as the authors reported, deconvolution methods underperform in estimating lowly abundant cell types. Therefore, conclusions from simulated mixtures based on compositions sampled from Uniform (0,1) are expected to be very limited in reliably ranking methods by their expected performance in real-world applications.

1. As mentioned higher, we disagree with the reviewer on the added value of limiting our analyses to real-world distributions. Using only real-world data makes it more

difficult to disentangle all the confounders acting in real-world scenarios, and will at best yield a conclusion on which method is best to deconvolute blood cell mixtures under physiological conditions. Indeed, independently replicated DNA methylation profiles of other purified cell types are unavailable.

2. We believe researchers are best helped by identifying and quantifying the various confounders in DNA methylation analyses. This indeed includes consideration of the purity of the reference profiles, technical variation, and the performance of algorithms in estimating either rare or abundant cell types. With this information in hand and considering the project background, researchers can make an informed decision on which algorithm is most appropriate for their question.
3. Furthermore, to alleviate the reviewer's concern, we have performed and included 3 analyses in which we perform deconvolution on real-life proportions distributions *in vitro*, *in silico* (deconvolution-based) and cytometry-quantified datasets.

Comment #1:

The authors added an evaluation of the R^2 metric per cell type, as I suggested. The authors added the JSD and an accuracy score they defined based on performance ranks. JSD and especially the rank-based evaluations are great additions to the benchmarking, however, the issue of sensitivity to cell-type abundance remains. All three metrics, R^2 , JSD, and RMSE are sensitive to cell-type abundance: if one or two cell types are, on average, much more abundant than the other cell types in the tissue (e.g., neutrophils in whole-blood mixtures), then the values in all three metrics, for a given method, will be dominated by the estimates of the abundant cell types. In whole-blood samples, for example, such benchmarking is likely to highlight the methods that are best at estimating neutrophils and not necessarily the methods that have the best overall performance (i.e., across cell types). While my concern is not an issue when sampling cell compositions from Uniform (0,1) (since there is no difference in abundance between cell types in this case), this might lead to severe biases when considering more realistic cell compositions, in which different cell types may drastically differ in their abundance.

We thank the reviewer for this comment, and for appreciating the additional evaluations included in the manuscript. We agree that this is an important point of concern. Therefore, we attempted to identify biases towards cell types for each performance metric, and this using realistic cell proportions estimated through deconvolution (Figure 1C, right panel). Although bias towards the least and most abundant cell types is clear for respectively RMSE and JSD values, this is not true for R^2 and (importantly) for the summarized accuracy score values (rebuttal figure 3). In conclusion, while some metrics are influenced by cell type abundance as suggested by the reviewer, others are not. The overall conclusions in our manuscript are centered on the R^2 and accuracy score. These conclusions are hence not severely biased towards performance for highly abundant cell types such as neutrophils in whole blood mixtures. Finally, we want to reiterate that researchers should opt for specific metrics to evaluate which algorithm is most suited to address their specific research questions, as already mentioned in the manuscript (Line 420-423).

Rebuttal figure 3: Dotplots showing summarized metrics (R^2 , root mean squared error, Jensen-Shannon divergence and accuracy score) per cell type, colored by mean abundance. Red diamonds show means of metrics over all cell types.

Comment

#5:

The authors' response is unrelated to my comment. I raised the question of whether reference data representing a specific population will be limited in the deconvolution of a different population. One example would be restricting the reference to include only aged individuals for deconvolving mixtures from young individuals (see my reference to Yousefi et al. 2015 as an example of why this question is of interest).

We thank the reviewer for this clarification. The reviewer's issue involves a long-standing problem in the field of DNA methylome analyses (and of biomedicine in general) that clearly cannot be resolved within our study. Indeed, reference samples do not necessarily accurately reflect study samples. Pertinent examples include differences in age, sex, ethnicity, smoking, drug or medication use, methodological differences such as sample procurement and storage, DNA extraction, DNA concentration and purity, and differences related to disease subtype and stage, and (for tumors) grade and mutational status. These and others can all confound analyses. Obviously, resolving this represents a major challenge for the field in the years and decades to come. A recommendation to maximally align test and study subject characteristics was added to the discussion (line 447-448).

To further highlight this issue, we now include the analysis suggested by the reviewer as a vignette to our benchmark: we compared deconvolution using references of individuals with an age that differs from the age of the mixtures. Specifically, we used samples from individuals below 30 years of age as reference, while the *in silico* mixtures were generated from samples older than 30 years (young vs old) and the other way around (old vs young). Unsurprisingly, all metrics performed more poorly, with median R^2 decreasing by 0.02, median RMSE increasing by 0.005, and JSD by 0.001. (rebuttal figure 4a). Relative ranking of deconvolution algorithms was however similar to a dataset without age discrepancy (rebuttal figure 4b). Hence, while a suboptimal reference reduces accuracy, it does not influence selection of a performant deconvolution algorithms. This analysis has now been added to the updated manuscript (Line 261-267).

Rebuttal figure 4: a) Scatterplot showing correlation between accuracy scores for algorithm-normalization combinations on *in silico* mixtures generated from a dataset with minimal technical variation compared to the reference (y-axis) and datasets with significant age difference compared to the reference (x-axis). b) Boxplots showing R^2 values for each deconvolution algorithm on *in silico* mixtures generated from a dataset with minimal technical variation compared to the reference and from a dataset with significant age difference compared to the reference.

Reviewer #2 (Remarks to the Author):

The authors have adequately addressed my comments. Please consider the paper for publication.

Reviewer #3 (Remarks to the Author):

My concerns have been addressed.

Reviewer #3 (Remarks on code availability):

Only demo data benchmark is provided on Github.

We thank the reviewer for this comment. We only provided demo data as this was required for publishing in *Nature Communications*, this however showcases the main workflow of the core deconvolution benchmarking.

Reviewer #1 (Remarks to the Author):

The authors adequately addressed my comments.